# Inhibitory control of correlated intrinsic variability in cortical networks

**Carsen Stringer[1†], Marius Pachitariu[2,3†], Nicholas A Steinmetz[2,4], Michael Okun[2‡], Peter Bartho[5], Kenneth D Harris[2,3], Maneesh Sahani[1], Nicholas A Lesica[6*]**

[1]Gatsby Computational Neuroscience Unit, University College London, London, United Kingdom; [2]Institute of Neurology, University College London, London, United Kingdom; [3]Department of Neuroscience, Physiology and Pharmacology, University College London, London, United Kingdom; [4]Institute of Ophthalmology, University College London, London, United Kingdom; [5]MTA TTK NAP B Sleep Oscillations Research Group, Budapest, Hungary; [6]Ear Institute, University College London, London, United Kingdom

**Abstract** Cortical networks exhibit intrinsic dynamics that drive coordinated, large-scale fluctuations across neuronal populations and create noise correlations that impact sensory coding. To investigate the network-level mechanisms that underlie these dynamics, we developed novel computational techniques to fit a deterministic spiking network model directly to multi-neuron recordings from different rodent species, sensory modalities, and behavioral states. The model generated correlated variability without external noise and accurately reproduced the diverse activity patterns in our recordings. Analysis of the model parameters suggested that differences in noise correlations across recordings were due primarily to differences in the strength of feedback inhibition. Further analysis of our recordings confirmed that putative inhibitory neurons were indeed more active during desynchronized cortical states with weak noise correlations. Our results demonstrate that network models with intrinsically-generated variability can accurately reproduce the activity patterns observed in multi-neuron recordings and suggest that inhibition modulates the interactions between intrinsic dynamics and sensory inputs to control the strength of noise correlations.

**\*For correspondence:** n.lesica@ucl.ac.uk

[†]These authors contributed equally to this work

**Present address:** [‡]Department of Neuroscience, Psychology and Behaviour, University of Leicester, Leicester, United Kingdom

**Competing interests:** The authors declare that no competing interests exist.

## Introduction

The patterns of cortical activity evoked by sensory stimuli provide the internal representation of the outside world that underlies perception. However, these patterns are driven not only by sensory inputs, but also by the intrinsic dynamics of the underlying cortical network. These dynamics can create correlations in the activity of neuronal populations with important consequences for coding and computation (*Shadlen et al., 1996*; *Abbott and Dayan, 1999*; *Averbeck et al., 2006*). The correlations between pairs of neurons have been studied extensively (*Cohen and Kohn, 2011*; *Ecker et al., 2010*; *Averbeck et al., 2006*), and recent studies have demonstrated that they are driven by dynamics involving coordinated, large-scale fluctuations in the activity of many cortical neurons (*Sakata and Harris, 2009*; *Pachitariu et al., 2015*; *Okun et al., 2015*). Inactivation of the cortical circuit suppresses these synchronized fluctuations at the level of the membrane potential, in both awake and anesthetized animals, suggesting that this synchronization is cortical in origin (*Cohen-Kashi Malina et al., 2016*). Importantly, the nature of these dynamics and the correlations that they create are dependent on the state of the underlying network; it has been shown that various factors modulate the strength of correlations, such as anesthesia (*Harris and Thiele, 2011*; *Schölvinck et al., 2015*; *Constantinople and Bruno, 2011*), attention (*Cohen and Maunsell, 2009*;

**eLife digest** Our brains contain billions of neurons, which are continually producing electrical signals to relay information around the brain. Yet most of our knowledge of how the brain works comes from studying the activity of one neuron at a time. Recently, studies of multiple neurons have shown that they tend to be active together in short bursts called "up" states, which are followed by periods in which they are less active called "down" states. When we are sleeping or under a general anesthetic, the neurons may be completely silent during down states, but when we are awake the difference in activity between the two states is usually less extreme. However, it is still not clear how the neurons generate these patterns of activity.

To address this question, Stringer et al. studied the activity of neurons in the brains of awake and anesthetized rats, mice and gerbils. The experiments recorded electrical activity from many neurons at the same time and found a wide range of different activity patterns. A computational model based on these data suggests that differences in the degree to which some neurons suppress the activity of other neurons may account for this variety. Increasing the strength of these inhibitory signals in the model decreased the fluctuations in electrical activity across entire areas of the brain. Further analysis of the experimental data supported the model's predictions by showing that inhibitory neurons – which act to reduce electrical activity in other neurons – were more active when there were fewer fluctuations in activity across the brain.

The next step following on from this work would be to develop ways to build computer models that can mimic the activity of many more neurons at the same time. The models could then be used to interpret the electrical activity produced by many different kinds of neuron. This will enable researchers to test more sophisticated hypotheses about how the brain works.

*Mitchell et al., 2009*; *Buran et al., 2014*), locomotion (*Schneider et al., 2014*; *Erisken et al., 2014*), and alertness (*Vinck et al., 2015*; *McGinley et al., 2015a*). In light of these findings, it is critical that we develop a deeper understanding of the origin and coding consequences of correlations at the biophysical network level.

While a number of modeling studies have explored the impact of correlations on sensory coding (*Shadlen et al., 1996*; *de la Rocha et al., 2007*; *Averbeck et al., 2006*; *Pillow et al., 2008*; *Ecker et al., 2011*; *Moreno-Bote et al., 2014*), there have been few efforts to identify their biophysical origin; the standard assumption that correlations arise from common input noise (*de la Rocha et al., 2007*; *Doiron et al., 2016*; *Lyamzin et al., 2015*) simply pushes the correlations from spiking to the membrane voltage without providing insight into their genesis. Models that use external noise to create correlations have been used in theoretical investigations of how network dynamics can transform correlations (*Doiron et al., 2016*), but no physiological source for the external noise used in these models has yet been identified. However, no external noise is needed to generate the correlated activity that is observed in vivo; in vitro experimental studies have shown that cortical networks are capable of generating large-scale fluctuations intrinsically (*Sanchez-Vives et al., 2010*; *Sanchez-Vives and McCormick, 2000*), and in vivo results suggest that the majority of cortical fluctuations arise locally (*Cohen-Kashi Malina et al., 2016*; *Shapcott et al., 2016*). If the major source of the correlations in cortical networks is, in fact, internal, then the network features that control these correlations may be different from those that control correlations in model networks with external noise.

We demonstrate that network models with intrinsic variability are indeed capable of reproducing the wide variety of activity patterns that are observed in vivo, and then proceed to use a large number of multi-neuron recordings and a model-based analysis to investigate the mechanisms that control intrinsically generated-noise correlations. For our results to provide direct insights into physiological mechanisms, we required a model with several properties: (1) the model must be able to internally generate the complex intrinsic dynamics of cortical networks, (2) it must be possible to fit the model parameters directly to spiking activity from individual multi-neuron recordings, and (3) the model must be biophysically interpretable and enable predictions that can be tested experimentally. No existing model satisfies all of these criteria; the only network models that have been fit directly to multi-neuron recordings have relied on either abstract dynamical systems (*Curto et al.,*

*2009*) or probabilistic frameworks in which variability is modelled as stochastic and correlated variability arises through abstract latent variables whose origin is assumed to lie either in unspecified circuit processes (*Ecker et al., 2014*; *Macke et al., 2011*; *Pachitariu et al., 2013*; *Pillow et al., 2008*) or elsewhere in the brain (*Goris et al., 2014*; *de la Rocha et al., 2007*). While these models are able to accurately reproduce many features of cortical activity and provide valuable summaries of the phenomenological and computational properties of cortical networks, their parameters are difficult to interpret at a biophysical level.

One alternative to these abstract stochastic models is a biophysical spiking network, (*van Vreeswijk and Sompolinsky, 1996*; *Amit and Brunel, 1997*; *Renart et al., 2010*; *Litwin-Kumar and Doiron, 2012*; *Wolf et al., 2014*). These networks can be designed to have interpretable parameters, but have not been shown to internally generate large-scale fluctuations and noise correlations of the kind routinely seen in multi-neuron recordings. Networks with structured connectivity have been shown to generate correlated activity in small groups containing less than 5% of all neurons (*Litwin-Kumar and Doiron, 2012*), but not in the entire network. Furthermore, large-scale neural network models have not yet been fit directly to multi-neuron recordings and, thus, their use has been limited to attempts to explain qualitative features of cortical dynamics through manual tuning of network parameters. This inability to fit the networks directly to recordings has made it difficult to identify which of these network features, if any, play an important role in vivo. To overcome this limitation, we used a novel computational approach that allowed us to fit spiking networks directly to individual multi-neuron recordings. By taking advantage of the computational power of graphics processing units (GPUs), we were able to simulate the network with millions of different parameter values for 900 seconds each to find those that best reproduced the structure of the activity in a given recording.

We developed a novel biophysical spiking network with intrinsic variability and a small number of parameters that was able to capture the apparently doubly chaotic structure of cortical activity (*Churchland and Abbott, 2012*). Previous models with intrinsic variability have been successful in capturing both the microscopic trial-to-trial variability in spike timing and long-timescale fluctuations in spike rate in individual neurons (*van Vreeswijk and Sompolinsky, 1996*; *Amit and Brunel, 1997*; *Vogels and Abbott, 2005*), but none of these models have been able to capture the coordinated, large-scale fluctuations that are shared across neurons. By combining spike-frequency adaptation (*Destexhe, 2009*; *Latham et al., 2000*) with high excitatory connectivity, our network is able to generate intrinsic global fluctuations that are of variable duration, arise at random times, and do not necessarily phase-lock to external input, thus creating noise correlations in evoked responses. This correlated intrinsic variability distinguishes our model from previous rate or spiking network models (*Parga and Abbott, 2007*; *Renart et al., 2010*; *Wolf et al., 2014*; *Doiron et al., 2016*), as well as from phenomenological dynamical systems (*Macke et al., 2011*; *Pachitariu et al., 2013*), all of which create noise correlations by injecting common noise into all neurons, an approach which, by construction, provides little insight into the biophysical mechanisms that generate the noise (*Doiron et al., 2016*).

To gain insight into the mechanisms that control noise correlations in vivo, we took the following approach: (1) we assembled multi-neuron recordings from different species, sensory modalities, and behavioral states to obtain a representative sample of cortical dynamics; (2) we generated activity from the network model to understand how each of its parameters controls its dynamics, and we verified that it was able to produce a variety of spike patterns that were qualitatively similar to those observed in vivo; (3) we fit the model network directly to the spontaneous activity in each of our recordings, and we verified that the spike patterns generated by the network quantitatively matched those in each recording; (4) we examined responses to sensory stimuli to determine which of the model parameters could account for the differences in noise correlations across recordings – the results of this analysis identified the strength of feedback inhibition as a key parameter and predicted that the activity of inhibitory interneurons should vary inversely with the strength of noise correlations; (5) we confirmed this prediction through additional analysis of our recordings showing that the activity of putative inhibitory neurons is increased during periods of cortical desynchronization with weak noise correlations in both awake and anesthetized animals; (6) we repeated all of the above analyses in recordings from mice during periods of locomotion to show that our results also apply to the cortical state transitions that are induced by natural behavior. Our results suggest that weak inhibition allows activity to be dominated by coordinated, large-scale fluctuations that cause

the state of the network to vary over time and, thus, create variability in the responses to successive stimuli that is correlated across neurons. In contrast, when inhibition is strong, these fluctuations are suppressed and the network state remains constant over time, allowing the network to respond reliably to successive stimuli and eliminating noise correlations.

## Results

### Cortical networks exhibit a wide variety of intrinsic dynamics

To obtain a representative sample of cortical activity patterns, we collected multi-neuron recordings from different species (mouse, gerbil, or rat), sensory modalities (A1 or V1), and behavioral states (awake or under one of several anesthetic agents). We compiled recordings from a total of 59 multi-neuron populations across six unique recording types (i.e. species/modality/state combinations; see *Supplementary file 1*). The spontaneous activity in different recordings exhibited striking differences not only in overall activity level, but also in the spatial and temporal structure of activity patterns; while concerted, large-scale fluctuations were prominent in some recordings, they were nearly absent in others (*Figure 1a*). In general, large-scale fluctuations were weak in awake animals and strong under anesthesia, but this was not always the case (see further examples in Figure 3 and summary statistics for each recording in *Figure 1—figure supplement 1*).

The magnitude and frequency of the large-scale fluctuations in each recording were reflected in the autocorrelation function of the multi-unit activity (MUA, the summed spiking of all neurons in the population in 15 ms time bins). The autocorrelation function of the MUA decayed quickly to zero for recordings with weak large-scale fluctuations, but had oscillations that decayed slowly for recordings with stronger fluctuations (*Figure 1b*). The activity patterns in recordings with strong large-scale fluctuations were characterized by clear transitions between up states, where most of the population was active, and down states, where the entire population was silent. These up and down state dynamics were reflected in the distribution of the MUA across time bins; recordings with strong large-scale fluctuations had a large percentage of time bins with zero spikes (*Figure 1c*).

To summarize the statistical structure of the activity patterns in each recording, we measured four quantities. We used mean spike rate to describe the overall level of activity, mean pairwise correlations to describe the spatial structure of the activity patterns, and two different measures to describe the temporal structure of the activity patterns – the decay time of the autocorrelation function of the MUA, and the percentage of MUA time bins with zero spikes. While there were some dependencies in the values of these quantities across different recordings (*Figure 1d*), there was also considerable scatter both within and across recording types. This scatter suggests that there is no single dimension in the space of cortical dynamics along which the overall level of activity and the spatial and temporal structure of the activity patterns all covary, but rather that cortical dynamics span a multidimensional continuum (*Harris and Thiele, 2011*). This was confirmed by principal component analysis; even in the already reduced space described by our summary statistics, three principal components were required to account for the differences in spike patterns across recordings (*Figure 1e*).

### A deterministic spiking network model of cortical activity

To investigate the network-level mechanisms that control cortical dynamics, we developed a biophysically-interpretable model that was capable of reproducing the wide range of activity patterns observed in vivo. We constructed a minimal deterministic network of excitatory spiking integrate-and-fire neurons with non-selective feedback inhibition and single-neuron adaptation currents (*Figure 2a*). Each neuron receives constant tonic input, and the neurons are connected randomly and sparsely with 5% probability. The neurons are also coupled indirectly through global, supralinear inhibitory feedback driven by the spiking of the entire network (*Rubin et al., 2015*), reflecting the near-complete interconnectivity between pyramidal neurons and interneurons in local populations (*Hofer et al., 2011*; *Fino and Yuste, 2011*; *Packer and Yuste, 2011*). The supralinearity of the inhibitory feedback is a critical feature of the network, as it shifts the balance of excitation and inhibition in favor of inhibition when the network is strongly driven, as has been observed in awake animals (*Haider et al., 2013*).

The model has five free parameters: three controlling the average strength of excitatory connectivity, the strength of inhibitory feedback, and the strength of adaptation, respectively, and two

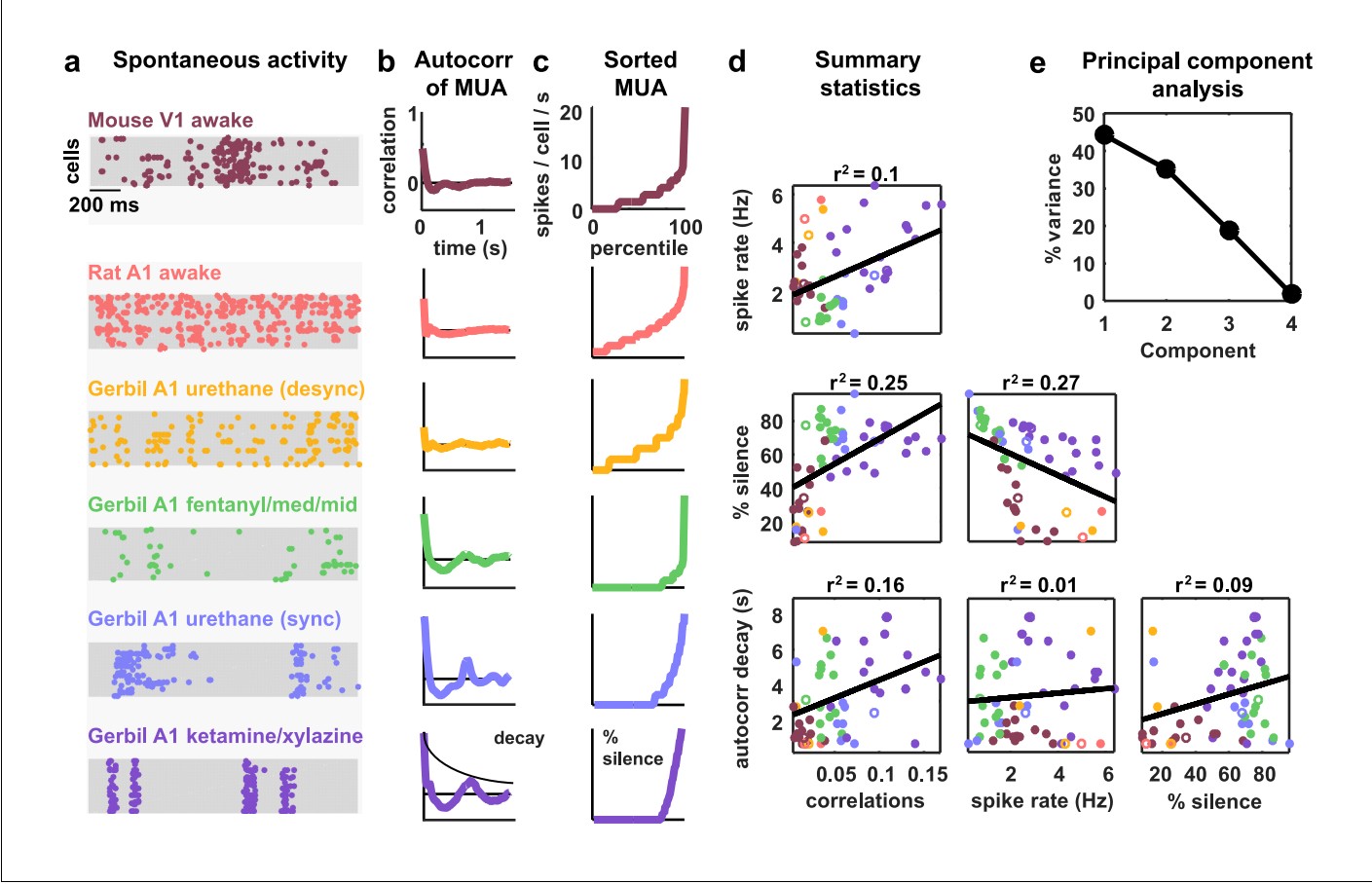

**Figure 1.** Cortical networks exhibit a wide variety of intrinsic dynamics. (a) Multi-neuron raster plots showing examples of a short segment of spontaneous activity from each of our recording types. Each row in each plot represents the spiking of one single unit. Note that recordings made under urethane were separated into two different recording types, synchronized (sync) and desynchronized (desync), as described in the Materials and methods. (b) The autocorrelation function of the multi-unit activity (MUA, the summed spiking of all neurons in the population in 15 ms time bins) for each example recording. The timescale of the autocorrelation function (the autocorr decay) was measured by fitting an exponential function to its envelope as indicated. (c) The values of the MUA across time bins sorted in ascending order. The percentage of time bins with zero spikes (the '% silence') is indicated. (d) Scatter plots showing all possible pairwise combinations of the summary statistics for each recording. Each point represents the values for one recording. Colors correspond to recording types as in (a). The recordings shown in (a) are denoted by open circles. The best fit line and the fraction of the variance that it explained are indicated on each plot. Spearman rank correlation p-values for each plot (from left to right, top to bottom) are as follows: $p<0.05, p<10^{-4}, p<10^{-5}, p<10^{-2}, p = 0.447, p<0.05$. (e) The percent of the variance in the summary statistics across recordings that is explained by each principal component of the values.

The following figure supplement is available for figure 1:

**Figure supplement 1.** Statistics for all fits.

controlling the strength of the tonic input to each neuron, which is chosen from an exponential distribution. The timescales that control the decay of the excitatory, inhibitory and adaptation currents are fixed at 5.10 ms, 3.75 ms and 375 ms, respectively. (These timescales have been chosen based on the physiologically known timescales of AMPA, GABA$_A$, and the calcium-dependent afterhyperpolarizing current. We also verified that the qualitative nature of our results did not change when we included slow conductances or clustered connectivity; see *Figure 2—figure supplement 1*.)

Note that no external noise input is required to generate variable activity; population-wide fluctuations over hundreds of milliseconds are generated when the slow adaptation currents synchronize across neurons to maintain a similar state of adaptation throughout the entire network, which, in turn, results in coordinated spiking (*Latham et al., 2000*; *Destexhe, 2009*). The variability in the

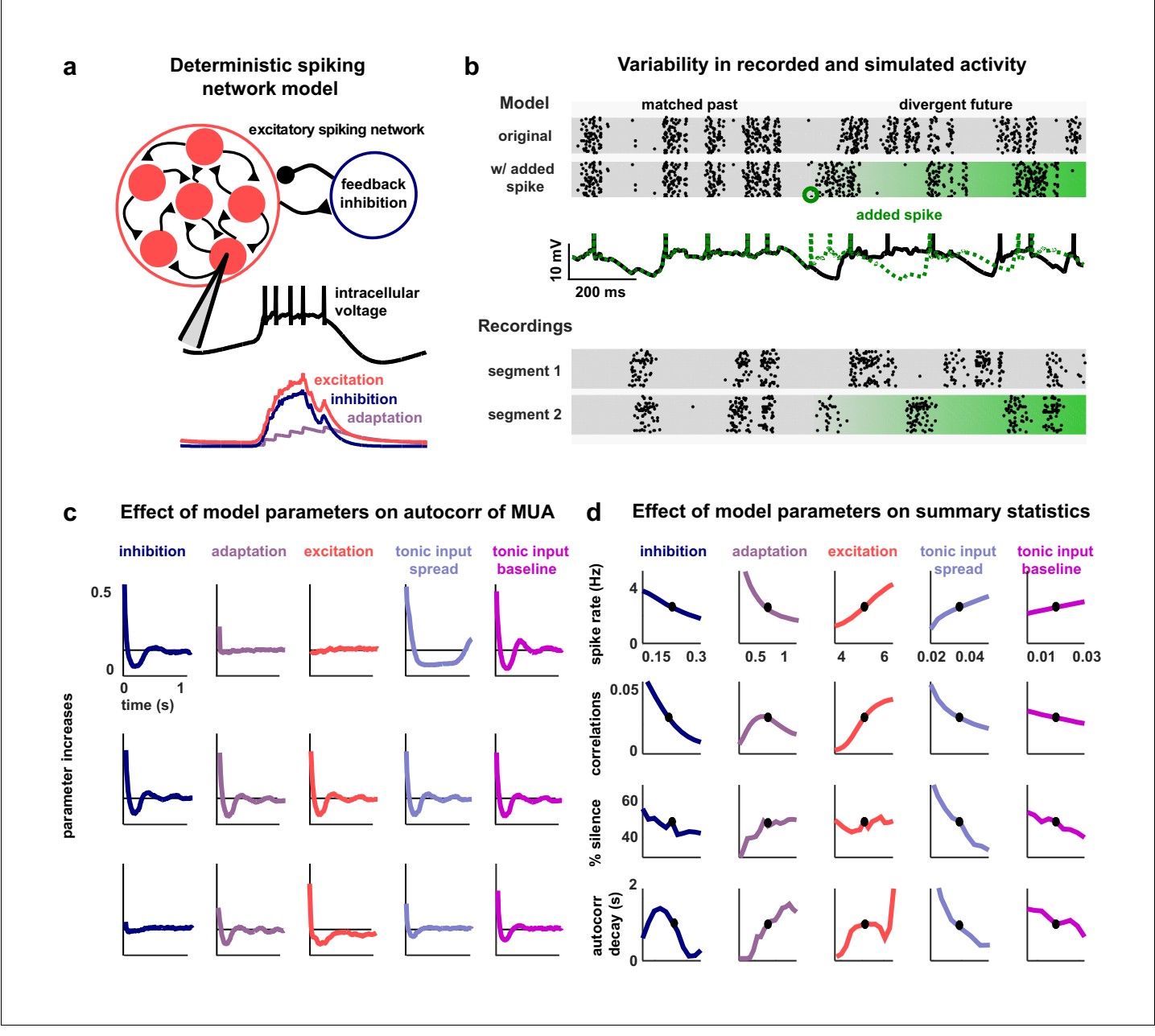

**Figure 2.** A deterministic spiking network model of cortical activity. (**a**) A schematic diagram of our deterministic spiking network model. An example of a short segment of the intracellular voltage of a model neuron is also shown, along with the corresponding excitatory, inhibitory and adaptation currents. (**b**) An example of macroscopic variability in cortical recordings and network simulations. The top two multi-neuron raster plots show spontaneous activity generated by the model. By adding a very small perturbation, in this case one spike added to a single neuron, the subsequent activity patterns of the network can change dramatically. The middle traces show the intracellular voltage of the model neuron to which the spike was added. The bottom two raster plots show a similar phenomenon observed in vivo. Two segments of activity extracted from different periods during the same recording were similar for three seconds, but then immediately diverged. (**c**) The autocorrelation function of the MUA measured from network simulations with different model parameter values. Each column shows the changes in the autocorrelation function as the value of one model parameter is changed while all others are held fixed. The fixed values used were $w_I = 0.22, w_A = 0.80, w_E = 4.50, b_1 = 0.03, b_0 = 0.013$. (**d**) The summary statistics measured from network simulations with different model parameter values. Each line shows the changes in the indicated summary statistic as one model parameter is changed while all others are held fixed. Fixed values were as in panel **c**.

The following figure supplement is available for figure 2:

**Figure supplement 1.** Model networks with long timescales and structured architecture.

model arises through chaotic amplification of small changes in initial conditions or small perturbations to the network that cause independent simulations to diverge. In some parameter regimes, the instability of the network is such that the structure of the spike patterns generated by the model is sensitive to changes in the spike times of individual neurons. In fact, a single spike added randomly to a single neuron during simulated activity is capable of changing the time course of large-scale fluctuations, in some cases triggering immediate population-wide spiking (*Figure 2b*, top rows). Similar phenomena have been observed in vivo previously (*London et al., 2010*) and were also evident in our recordings when comparing different extracts of cortical activity; spike patterns that were similar for several seconds often then began to diverge almost immediately (*Figure 2b*, bottom rows).

## Multiple features of the network model can control its dynamics

The dynamical regime of the network model is determined by the interactions between its different features. To determine the degree to which each feature of the network was capable of influencing the structure of its activity patterns, we analyzed the effects of varying the value of each model parameter. We started from a fixed set of parameter values and simulated activity while independently sweeping each parameter across a wide range of values. The results of these parameter sweeps clearly demonstrate that each of the five parameters can exert strong control over the dynamics of the network, as both the overall level of activity and the spatial and temporal structure of the patterns in simulated activity varied widely with changes in each parameter (*Figure 2c–d*).

With the set of fixed parameter values used for the parameter sweeps, the network is in a regime with slow, ongoing fluctuations between up and down states. In this regime, the amplification of a small perturbation results in a sustained, prolonged burst of activity (up state), which, in turn, drives a build-up of adaptation currents that ultimately silences the network for hundreds of milliseconds (down state) until the cycle repeats. These fluctuations can be suppressed by an increase in the strength of feedback inhibition, which eliminates slow fluctuations and shifts the network into a regime with weak, tonic spiking and weak correlations (*Figure 2c–d*, first column); in this regime, small perturbations are immediately offset by the strong inhibition and activity is returned to baseline. Strong inhibition also offsets externally-induced perturbations in balanced networks (*Renart et al., 2010*), but in our model such perturbations are internally-generated and would result in runaway excitation in the absence of inhibitory stabilization. The fluctuations between up and down states can also be suppressed by decreasing adaptation (*Figure 2c–d*, second column); without adaptation currents to create slow, synchronous fluctuations across the network, neurons exhibit strong, tonic spiking.

The dynamics of the network can also be influenced by changes in the strength of the recurrent excitation or tonic input. Increasing the strength of excitation results in increased activity and stronger fluctuations, as inhibition is unable to compensate for the increased amplification of small perturbations (*Figure 2c–d*, third column). Increasing the spread or baseline level of tonic input also results in increased activity, but with suppression, rather than enhancement, of slow fluctuations (*Figure 2c–d*, fourth and fifth column). As either the spread or baseline level of tonic input is increased, more neurons begin to receive tonic input that is sufficient to overcome their adaptation current and, thus, begin to quickly reinitiate up states after only brief down states and, eventually, transition to tonic spiking.

## The network model reproduces the dynamics observed in vivo

The network simulations demonstrate that each of its features is capable of controlling its dynamics and shaping the structure of its activity patterns. To gain insight into the mechanisms that may be responsible for creating the differences in dynamics observed in vivo, we fit the model to each of our recordings. We optimized the model parameters so that the patterns of activity generated by the network matched those observed in spontaneous activity (*Figure 3a*). We measured the agreement between the simulated and recorded activity by a cost function which was the sum of discrepancies in the autocorrelation function of the MUA, the distribution of MUA values across time bins, and the mean pairwise correlations. Together, these statistics describe the overall level of activity in each recording, as well as the spatial and temporal structure of its activity patterns.

Fitting the model to the recordings required us to develop new computational techniques. The network parametrization is fundamentally nonlinear, and the statistics used in the cost function are

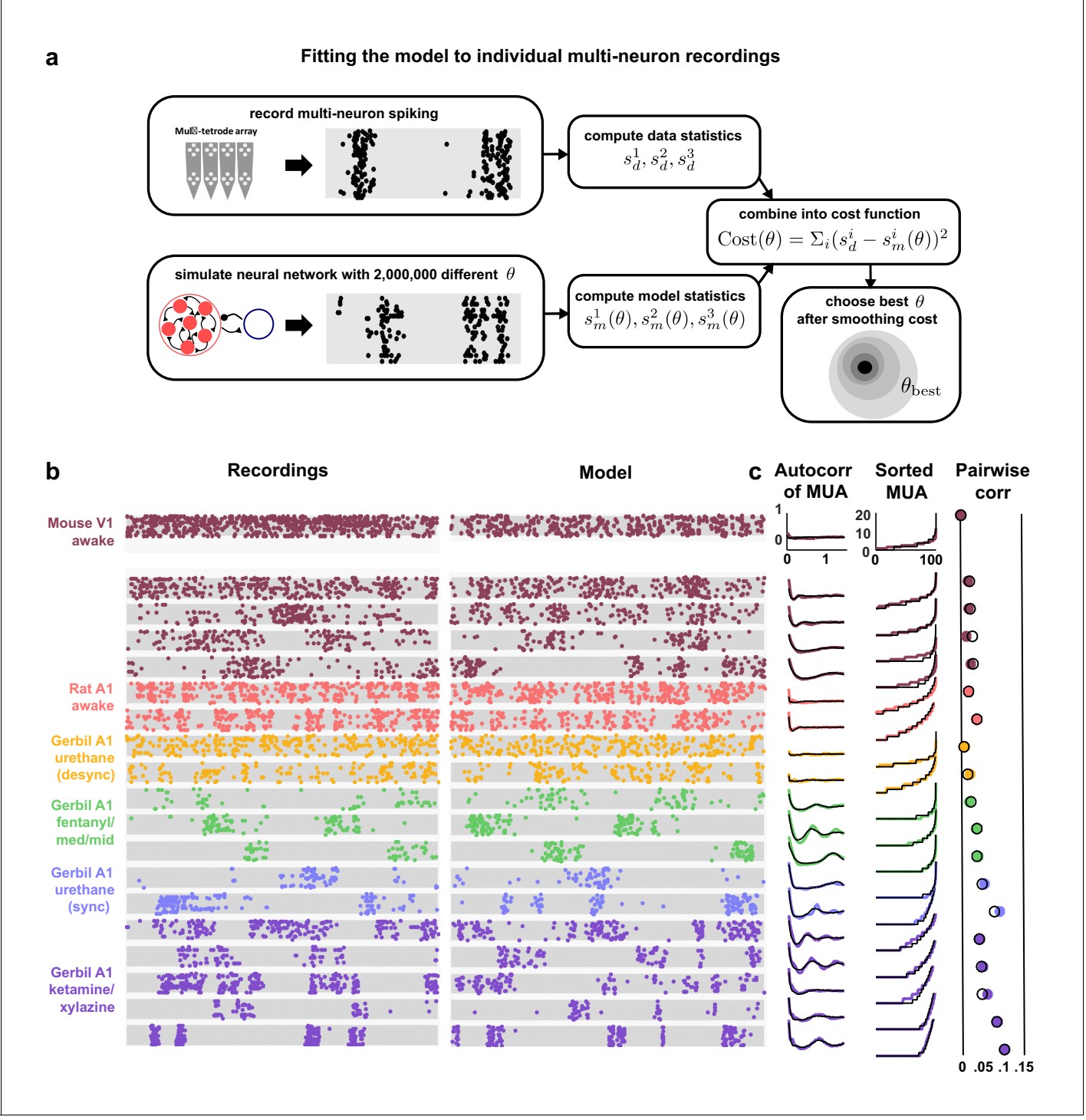

**Figure 3.** Deterministic spiking networks reproduce the dynamics observed in vivo. (a) A schematic diagram illustrating how the parameters of the network model were fit to individual multi-neuron recordings. (b) Examples of spontaneous activity from different recordings, along with spontaneous activity generated by the model fit to each recording. (c) The left column shows the autocorrelation function of the MUA for each recording, plotted as in *Figure 1*. The black lines show the autocorrelation function measured from spontaneous activity generated by the model fit to each recording. The middle column shows the sorted MUA for each recording along with the corresponding model fit. The right column shows the mean pairwise correlations between the spiking activity of all pairs of neurons in each recording (after binning activity in 15 ms bins). The colored circles show the correlations measured from the recordings and the black open circles show the correlations measured from spontaneous activity generated by the model fit to each recording.

*Figure 3 continued on next page*

*Figure 3 continued*

The following figure supplements are available for figure 3:

**Figure supplement 1.** Optimization performance of the MCMC procedure.

**Figure supplement 2.** Costs and parameter fits.

**Figure supplement 3.** Variance explained by model fits.

**Figure supplement 4.** Analysis of local minima.

themselves nonlinear functions of a dynamical system with discontinuous integrate-and-fire mechanisms. Thus, as no gradient information was available to guide the optimization, we used Monte Carlo simulations to generate activity and measure the relevant statistics with different parameter values. By using GPU computing resources, we were able to design and implement network simulations that ran 10000x faster than real time, making it feasible to sample the cost function with high resolution and locate its global minimum to identify the parameter configuration that resulted in activity patterns that best matched those of each recording. We also verified that the global minimum of the cost function could be identified with 10x fewer samples of simulated activity using a Gibbs sampling optimizer with simulated annealing (*Figure 3—figure supplement 1*), but the results presented below are based on the global minima identified by the complete sampling of parameter space.

The model was flexible enough to capture the wide variety of activity patterns observed across our recordings, producing both decorrelated, tonic spiking and coordinated, large-scale fluctuations between up and down states as needed (see examples in *Figure 3b*, statistics for all recordings and models in *Figure 1—figure supplement 1*, and parameter values and goodness-of-fit measures for all recordings in *Figure 3—figure supplement 2*). The fits were also quantitatively accurate. We found that the median variance explained by the model of the autocorrelation function of the MUA, the distribution of MUA values across time bins, and the mean pairwise correlations were 82%, 90%, and 97% respectively (*Figure 3—figure supplement 3b*). In fact, these fits were about as good as possible given the length of our recordings: the fraction of the variance in the statistics of one half of each recording that was explained by the statistics of the other half of the recording were 84%, 98%, and 100% respectively (*Figure 3—figure supplement 3a*). Because we used a cost function that captured many different properties of the recorded activity while fitting only a very small number of model parameters, the risk of network degeneracies was relatively low (*Gutierrez et al., 2013*; *Marder et al., 2015*). Nonetheless, we also confirmed that analysis of model parameters corresponding to local minima of the cost function did not lead to a different interpretation of our results (see *Figure 3—figure supplement 4*).

## Strong inhibition suppresses noise correlations

Our main interest was in understanding how the different network-level mechanisms that are capable of controlling intrinsic dynamics contribute to the correlated variability in responses evoked by sensory stimuli. The wide variety of intrinsic dynamics in our recordings was reflected in the differences in evoked responses across recording types; while some recordings contained strong, reliable responses to the onset of a stimulus, other recordings contained responses that were highly variable across trials (*Figure 4a*). There were also large differences in the extent to which the variability in evoked responses was correlated across the neurons in each recording; pairwise noise correlations were large in some recordings and extremely weak in others, even when firing rates were similar (*Figure 4b*).

Because evoked spike patterns can depend strongly on the specifics of the sensory stimulus, we could not make direct comparisons between experimental responses across different species and modalities; our goal was to identify the internal mechanisms that are responsible for the differences in noise correlations across recordings and, thus, any differences in spike patterns due to differences in external input would confound our analysis. To overcome this confound and enable the

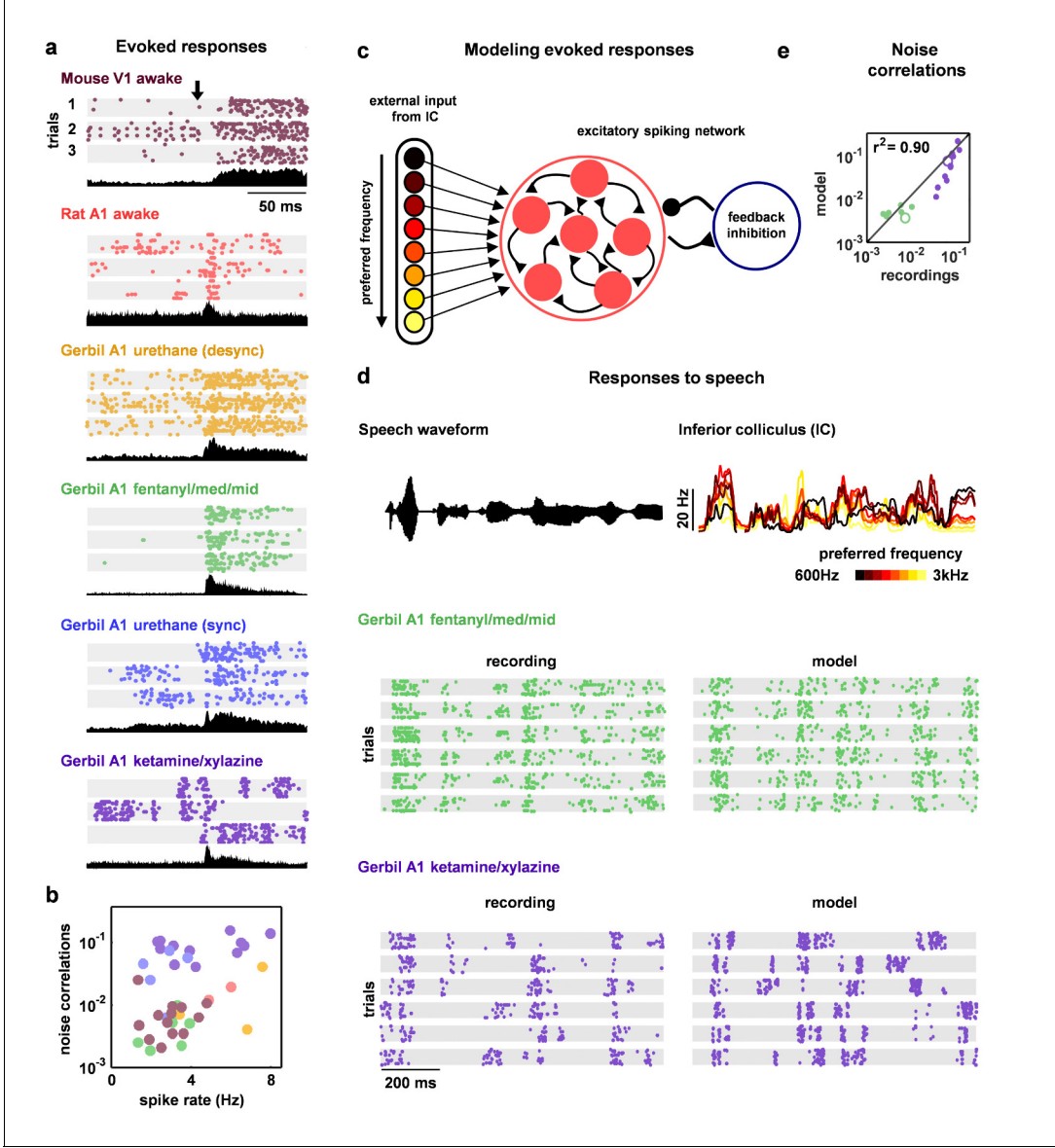

**Figure 4.** Deterministic spiking networks reproduce the noise correlations observed in vivo. (**a**) Multi-neuron raster plots and PSTHs showing examples of evoked responses from each of our recording types. Each row in each raster plot represents the spiking of one single unit. Each raster plot for each recording type shows the response on a single trial. The PSTH shows the MUA averaged across all presentations of the stimulus. Different stimuli were used for different recording types (see Materials and methods). (**b**) A scatter plot showing the mean spike rates and mean pairwise noise correlations (after binning the evoked responses in 15 ms bins) for each recording. Each point represents the values for one recording. Colors correspond to recording types as in (**a**). Values are only shown for the 38 of 59 recordings that contained both spontaneous activity and evoked responses. The Spearman's rank correlation was significant with p=0.0105. (**c**) A schematic diagram illustrating the modelling of evoked responses. We constructed the external input using recordings of responses from more than 500 neurons in the inferior colliculus (IC), the primary relay nucleus of the auditory midbrain that provides the main input to the thalamocortical circuit. We have shown previously that the Fano factors of the responses of IC neurons are close to one and the noise correlations between neurons are extremely weak (*Garcia-Lazaro et al., 2013*), suggesting that the spiking activity of a population of IC neurons can be well described by series of independent, inhomogeneous Poisson processes. To generate the responses of each model network to the external input, we averaged the activity of each IC neuron across trials, grouped the IC neurons by their preferred frequency, and selected a randomly chosen subset of 10 neurons from the same frequency group to drive each cortical neuron. (**d**) The top left plot shows the sound waveform presented in the IC recordings used as input to the model cortical network. The top right plot shows PSTHs formed by averaging IC responses across trials and across all IC neurons in each preferred frequency group. The raster plots show the recorded responses of two cortical populations on successive trials, along with the activity generated by the network model fit to each recording when driven by IC responses to the same sounds. (**e**) A scatter plot showing the noise correlations of responses measured from the actual recordings and from simulations of the network model fit to each recording when driven by IC responses to the same sounds. The Spearman rank correlation for the recordings versus the model were p<$10^{-5}$. The recordings shown in (**d**) are denoted by open circles.

*Figure 4 continued on next page*

*Figure 4 continued*

The following figure supplement is available for figure 4:

**Figure supplement 1.** Parameter sweeps for responses to external input.

comparison of noise correlations across recording types, we simulated the response of the network to the same external input for all recordings. We constructed the external input using recordings of spiking activity from the inferior colliculus (IC), a primary relay nucleus in the subcortical auditory pathway (*Figure 4c–d*). Using the subset of our cortical recordings in which we presented the same sounds that were also presented during the IC recordings, we verified that the noise correlations in the simulated cortical responses were similar to those in the recordings (*Figure 4e*).

The parameter sweeps described in *Figure 2* demonstrated that there are multiple features of the model network that can control its intrinsic dynamics, and a similar analysis of the noise correlations in simulated responses to external input produced similar results (*Figure 4—figure supplement 1*). To gain insight into which of these features could account for the differences in noise correlations across our recordings, we examined the dependence of the strength of the noise correlations in each recording on each of the model parameters. While several parameters were able to explain a significant amount of the variance in noise correlations across recordings, the amount of variance explained by the strength of inhibitory feedback was by far the largest (*Figure 5a*). The predominance of inhibition in the control of noise correlations was confirmed by the measurement of partial correlations (the correlation between the noise correlations and each parameter that remains after factoring out the influence of the other parameters; partial $r^2$ for inhibition: 0.67, excitation: 0.02, adaptation: 0.08, tonic input spread: 0.17, and tonic input baseline: 0.04). We also performed parameter sweeps to confirm that varying only the strength of inhibition was sufficient to result in large changes in noise correlations in the parameter regime of each recording (*Figure 5b*).

## Strong inhibition sharpens tuning and enables accurate decoding

We also examined how different features of the network controlled other aspects of evoked responses. We began by examining the extent to which differences in the value of each model parameter could explain differences in stimulus selectivity across recordings. To estimate selectivity, we drove the model network that was fit to each cortical recording with external inputs constructed from IC responses to tones, and used the simulated responses to measure the width of the frequency tuning curves of each model neuron. Although each model network received the same external inputs, the selectivity of the neurons in the different networks varied widely. The average tuning width of the neurons in each network varied most strongly with the strength of the inhibitory feedback in the network (*Figure 5c*; partial $r^2$ for inhibition: 0.74, excitation: 0.06, adaptation: 0.48, tonic input spread: 0.01, and tonic input baseline: 0.37), and varying the strength of inhibition alone was sufficient to drive large changes in tuning width (*Figure 5d*). These results are consistent with experiments demonstrating that inhibition can control the selectivity of cortical neurons (*Lee et al., 2012*), but suggest that this control does not require structured lateral inhibition.

We also investigated the degree to which the activity patterns generated by the model fit to each cortical recording could be used to discriminate different external inputs. We trained a decoder to infer which of seven possible stimuli evoked a given single-trial activity pattern and examined the extent to which differences in the value of each model parameter could account for the differences in decoder performance across recordings. Again, the amount of variance explained by the strength of inhibitory feedback was by far the largest (*Figure 5e*; partial $r^2$ for inhibition: 0.5, excitation: 0.16, adaptation 0.27, tonic input spread 0.02, and tonic input baseline 0.03); decoding was most accurate for activity patterns generated by networks with strong inhibition, consistent with the weak noise correlations and high selectivity of these networks. Parameter sweeps confirmed that varying only the strength of inhibition was sufficient to result in large changes in decoder performance (*Figure 5f*).

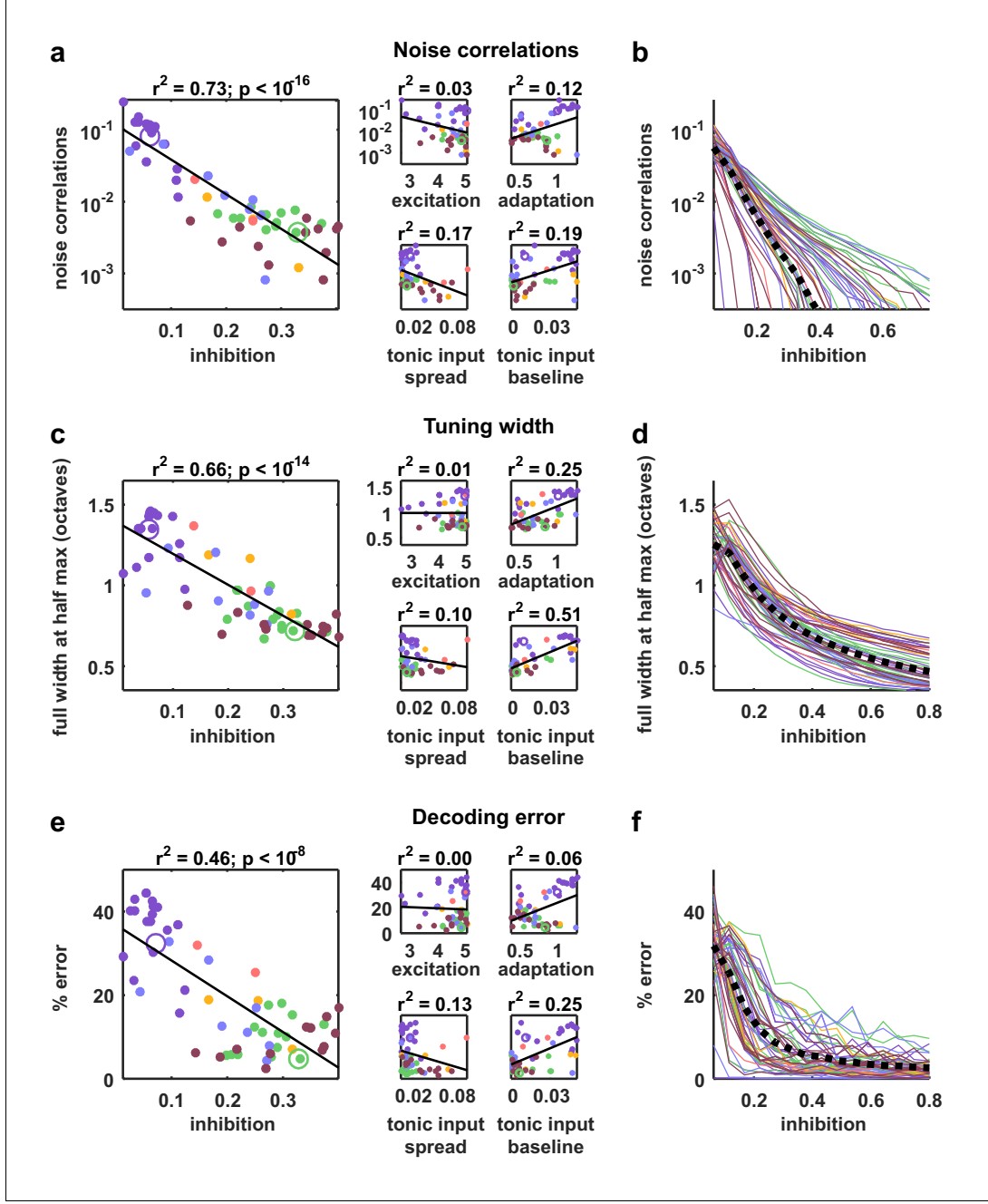

**Figure 5.** Strong inhibition suppresses noise correlations and enhances selectivity and decoding. (a) Scatter plots showing the mean pairwise noise correlations measured from simulations of the network model fit to each recording when driven by external input versus the value of the different model parameters. Colors correspond to recording types as in *Figure 4*. The recordings shown in *Figure 4d* are denoted by open circles. Spearman's rank correlation p-values for inhibition, excitation, adaptation, tonic input spread, and tonic input baseline were $p < 10^{-18}$, $p = 0.339$, $p = 0.011$, $p < 10^{-2}$, and $p < 10^{-3}$ respectively. (b) The mean pairwise noise correlations measured from network simulations with different values of the inhibition parameter $w_I$. The values of all other parameters were held fixed at those fit to each recording. Each line corresponds to one recording. Colors correspond to recording types as in *Figure 4*. (c,e) Scatter plots showing tuning width and decoding error, plotted as in (a). For (c), Spearman rank correlation p-values for inhibition, excitation, adaptation, tonic input spread, and tonic input baseline were $p < 10^{-15}$, $p = 0.642$, $p < 10^{-4}$, $p < 10^{-2}$, and $p < 10^{-9}$ respectively. For (e), Spearman rank correlation p-values for inhibition, excitation, adaptation, tonic input spread, and tonic input baseline were

*Figure 5 continued on next page*

*Figure 5 continued*

$p<10^{-9}$, $p=0.799$, $p=0.0766$, $p<10^{-2}$, and $p<10^{-4}$ respectively. (d,f) The tuning width and decoding error measured from network simulations with different values of the inhibition parameter $w_I$, plotted as in (b).

## Activity of fast-spiking (FS) neurons is increased during periods of cortical desynchronization with weak noise correlations

Our model-based analyses suggest an important role for feedback inhibition in controlling the way in which responses to sensory inputs are shaped by intrinsic dynamics. In particular, our results predict that inhibition should be strong in dynamical regimes with weak noise correlations. To test this prediction, we performed further analysis of our recordings to estimate the strength of inhibition in each recorded population. We classified the neurons in each recording based on the width of their spike waveforms (*Figure 6—figure supplement 1*). The waveforms for all recording types fell into two distinct clusters, allowing us to separate fast-spiking (FS) neurons from regular-spiking (RS) neurons. In general, more than 90% of FS cortical neurons have been reported to be parvalbumin-positive (PV+) inhibitory neurons (*Nowak et al., 2003*; *Kawaguchi and Kubota, 1997*; *Barthó et al., 2004*; *Cho et al., 2010*; *Madisen et al., 2012*; *Stark et al., 2013*; *Cohen and Mizrahi, 2015*), and this value approaches 100% in the deep cortical layers where we recorded (*Cardin et al., 2009*). While the separation of putative inhibitory and excitatory neurons based on spike waveforms is imperfect (nearly all FS neurons are inhibitory, but a small fraction (less than 20%) of RS neurons are also inhibitory [*Markram et al., 2004*]), it is still effective for approximating the overall levels of inhibitory and excitatory activity in a population.

Given the results of our model-based analyses, we hypothesized that the overall level of activity of FS neurons should vary inversely with the strength of noise correlations. To identify sets of trials in each recording that were likely to have either strong or weak noise correlations, we measured the level of cortical synchronization. Previous studies have shown that noise correlations are strong when the cortex is in a synchronized state, where activity is dominated by concerted, large-scale fluctuations, and weak when the cortex is in a desynchronized state, where these fluctuations are suppressed (*Pachitariu et al., 2015*; *Schölvinck et al., 2015*).

We began by analyzing our recordings from V1 of awake mice. We classified the cortical state during each stimulus presentation based on the ratio of low-frequency LFP power to high-frequency LFP power (*Sakata and Harris, 2012*) and compared evoked responses across the most synchronized and desynchronized subsets of trials (*Figure 6a*). As expected, noise correlations were generally stronger during synchronized trials than during desynchronized trials, and this variation in noise correlations with cortical synchrony was evident both within individual recordings and across animals (*Figure 6b–c*). As predicted by our model-based analyses, the change in noise correlations with cortical synchrony was accompanied by a change in FS activity; there was a four-fold increase in the mean spike rate of FS neurons from the most synchronized trials to the most desynchronized trials, while RS activity remained constant (*Figure 6d–f*).

We next examined our recordings from gerbil A1 under urethane in which the cortex exhibited transitions between distinct, sustained synchronized and desynchronized states (*Figure 6g*). As in our awake recordings, cortical desynchronization under urethane was accompanied by a decrease in noise correlations and an increase in FS activity (*Figure 6h–k*). In fact, both FS and RS activity increased with cortical desynchronization under urethane, but the increase in FS activity was much larger (110% and 42%, respectively). The increase in RS activity suggests that cortical desynchronization under urethane may involve other mechanisms in addition to an increase in feedback inhibition (a comparison of the model parameters fit to desynchronized and synchronized urethane recordings (*Figure 3—figure supplement 2*) suggests that the average level of tonic input is significantly higher during desynchronization (desynchronized: $0.075 \pm 0.008$, synchronized: $0.0195 \pm 0.0054$, $p = 0.006$)).

## The change in cortical state that accompanies locomotion can be explained by an increase in feedback inhibition

Finally, we asked whether the same mechanisms might be used to control the changes in network dynamics that accompany transitions in behavioral state, such as those induced by locomotion. We

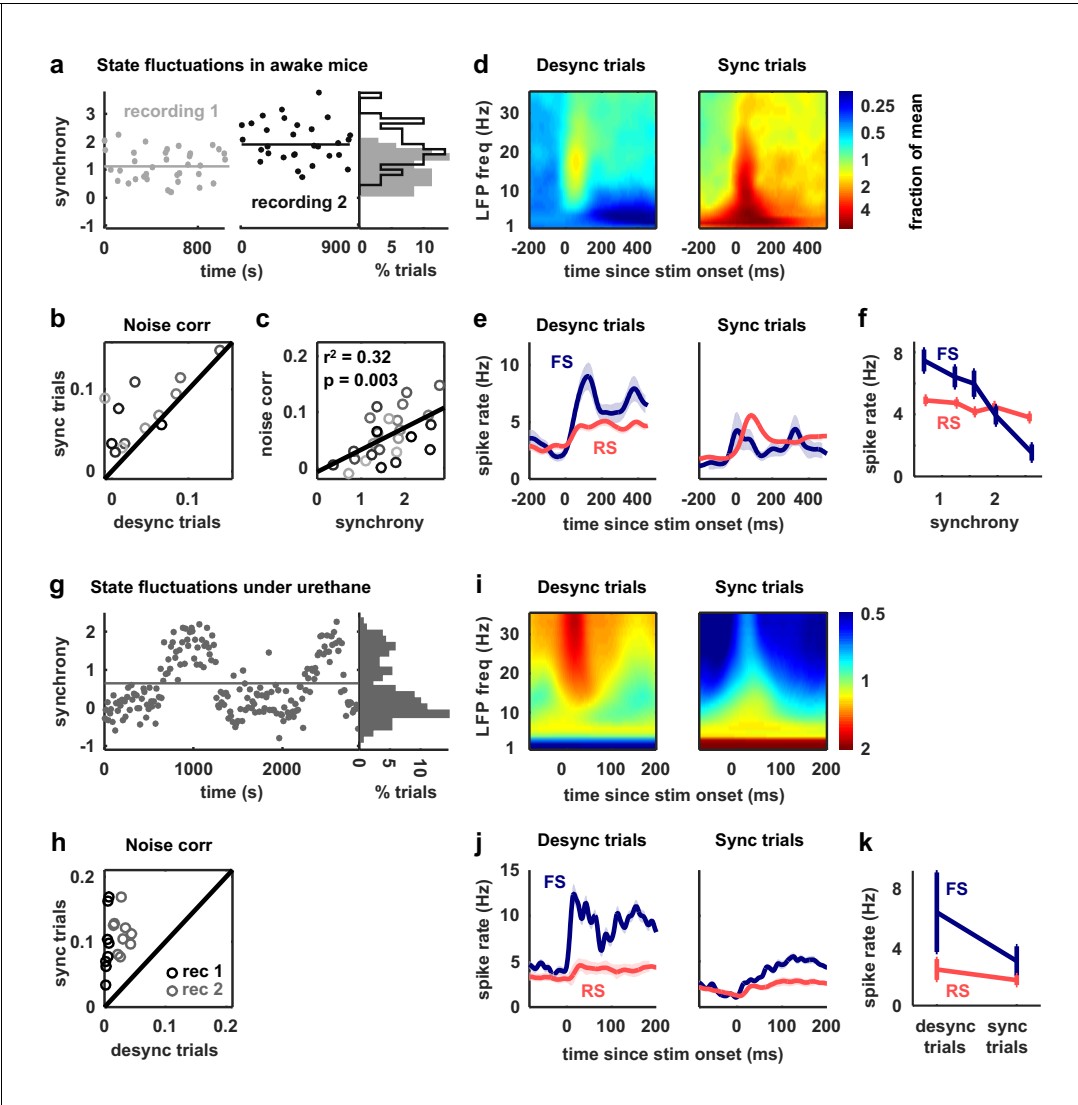

**Figure 6.** Fast-spiking neurons are more active during periods of cortical desynchronization with weak noise correlations. (**a**) The cortical synchrony at different points during two recordings from V1 of awake mice, measured as the log of the ratio of low-frequency (3–10 Hz) LFP power to high-frequency (11–96 Hz). The distribution of synchrony values across each recording is also shown. The lines indicate the median of each distribution. (**b**) A scatter plot showing the noise correlations measured during trials in which the cortex was in either a relatively synchronized (sync) or desynchronized (desync) state for each recording. Each point indicates the mean pairwise correlations between the spiking activity of all pairs of neurons in one recording (after binning the activity in 15 ms bins). Trials with the highest 50% of synchrony values were classified as sync and trials with the lowest 50% of synchrony values were classified as desync. Values for 13 different recordings are shown. The Wilcoxon two-sided signed-rank test p-value was $p<10^{-2}$. (**c**) A scatter plot showing noise correlations versus the mean synchrony for trials with the highest and lowest 50% of synchrony values for each recording. Colors indicate different recordings. The Spearman rank correlation significance among all recordings was $p<10^{-2}$. (**d**) Spectrograms showing the average LFP power during trials with the highest (sync) and lowest (desync) 20% of synchrony values across all recordings. The values shown are the deviation from the average spectrogram computed over all trials. (**e**) The average PSTHs of FS and RS neurons measured from evoked responses during trials with the highest (sync) and lowest (desync) 20% of synchrony values across all recordings. The lines show the mean across all neurons, and the error bars indicate ±1 SEM. (**f**) The median spike rate of FS and RS neurons during the period from 0 to 500 ms following stimulus onset, averaged across trials in each synchrony quintile. The lines show the mean across all neurons, and the error bars indicate ±1 SEM. The Wilcoxon two-sided signed-rank test comparing FS activity between the highest and lowest quintile had a significance of $p<10^{-9}$ and for RS activity, the significance was $p<10^{-2}$. (**g**) The cortical synchrony at different points during a urethane recording, plotted as in (**a**). The line indicates the value used to classify trials as synchronized (sync) or desynchronized (desync). (**h**) A scatter plot showing the noise correlations measured during trials in which the cortex was in either a synchronized (sync) or desynchronized (desync) state. Values for two different recordings are shown. Each point for each recording shows the noise correlations measured from responses to a different sound. The Wilcoxon two-sided signed-rank test between sync and desync state noise correlations had a significance of $p<10^{-3}$. (**i**) Spectrograms showing the average LFP power during synchronized and desynchronized trials, plotted as in (**d**). (**j**) The average PSTHs of FS and RS neurons during synchronized and desynchronized trials, plotted as in (**e**). (**k**) The median spike rate of FS and regular-

*Figure 6 continued on next page*

*Figure 6 continued*

spiking RS neurons during the period from 0 to 500 ms following stimulus onset during synchronized and desynchronized trials. The points show the mean across all neurons, and the error bars indicate ±1 SEM. The Wilcoxon two-sided signed-rank test comparing FS activity between the sync and desync had a significance of $p<10^{-3}$ and for RS activity, the significance was $p<10^{-5}$.

The following figure supplement is available for figure 6:

**Figure supplement 1.** Classification of neuron types by spike width.

recorded four separate populations of 100–200 neurons each, from two head-fixed mice that were allowed to run on a treadmill. We found that stationary periods were often accompanied by slow timescale population-wide fluctuations in firing (*Figure 7a–b*, top row). We fit the network model to these stationary periods, and verified that we could reproduce these dynamics (*Figure 7a–b*, top row, and statistics for all recordings and models in *Figure 7—figure supplement 1*). Running epochs were, by comparison, much more desynchronized (*Figure 7a–b*, bottom row), consistent with previous observations made with intracellular and LFP measurements (*Vinck et al., 2015*; *Niell and Stryker, 2010*; *McGinley et al., 2015a*; *Polack et al., 2013*; *Bennett et al., 2013*).

To determine which changes in our model best captured this state transition, we allowed either one or two parameters to change from the values fit to stationary periods. By changing two parameters, inhibition and adaptation, the model was able to reproduce the statistics of the neural population activity during running (*Figure 7a–b*, bottom row). Out of all the possible single-parameter changes, the best fits were achieved through changes in inhibition, while out of all the possible two-parameter changes, the best fits were achieved through changes in inhibition and adaptation (*Figure 7c*). In all four recordings, the model captured the change in dynamics associated with running through an increase in inhibition and a decrease in adaptation (*Figure 7d*). The changes in FS and RS activity in the recordings were consistent with such changes. Although both FS and RS populations increased their activity during running, the relative increase in FS activity was significantly larger (*Figure 7e*; on average, FS activity increased by 87% and RS activity increased by 28%). Our results suggest that the increase in RS activity during running despite increased FS activity is likely due to an accompanying decrease in adaptation.

## Discussion

We have shown here that a deterministic spiking network model is capable of intrinsically generating population-wide fluctuations in neural activity, without requiring external modulating inputs. It has been observed in vitro that population-wide fluctuations in neural activity persist without external input (*Sanchez-Vives et al., 2010*; *Sanchez-Vives and McCormick, 2000*). Such fluctuations also arise in vivo in localized cortical networks, in both awake and anesthetized animals, without feedforward inputs (*Shapcott et al., 2016*) or any external inputs (*Cohen-Kashi Malina et al., 2016*). However, no previous models have been able to reproduce such large-scale coordinated activity in a deterministic network of connected neurons; previous models only reproduced single-neuron variability (*Vogels and Abbott, 2005*; *Litwin-Kumar and Doiron, 2012*). By fitting our spiking network model with adaptation currents directly to experimental recordings, we demonstrated that the model is able to reproduce the wide variety of multi-neuron cortical activity patterns observed in vivo without the need for external noise. Through chaotic amplification of small perturbations, the model generates activity with both trial-to-trial variability in the spike times of individual neurons and coordinated, large-scale fluctuations of the entire network. These fluctuations continue in the presence of sensory stimulation, thus creating noise correlations in a deterministic neural network.

The development of a network model that can reproduce experimentally-observed activity patterns through intrinsic variability alone is a major advance beyond previous models (*Doiron et al., 2016*; *de la Rocha et al., 2007*; *Renart et al., 2010*; *Ecker et al., 2014*). Networks in the classical balanced state produce activity with zero mean pairwise correlations between neurons (*Doiron et al., 2016*; *van Vreeswijk and Sompolinsky, 1996*; *Renart et al., 2010*) and, thus, are not suitable to describe the population-wide fluctuations that are observed in many brain states in vivo (*Okun et al., 2015*). To obtain single-neuron rate fluctuations in balanced networks, structured

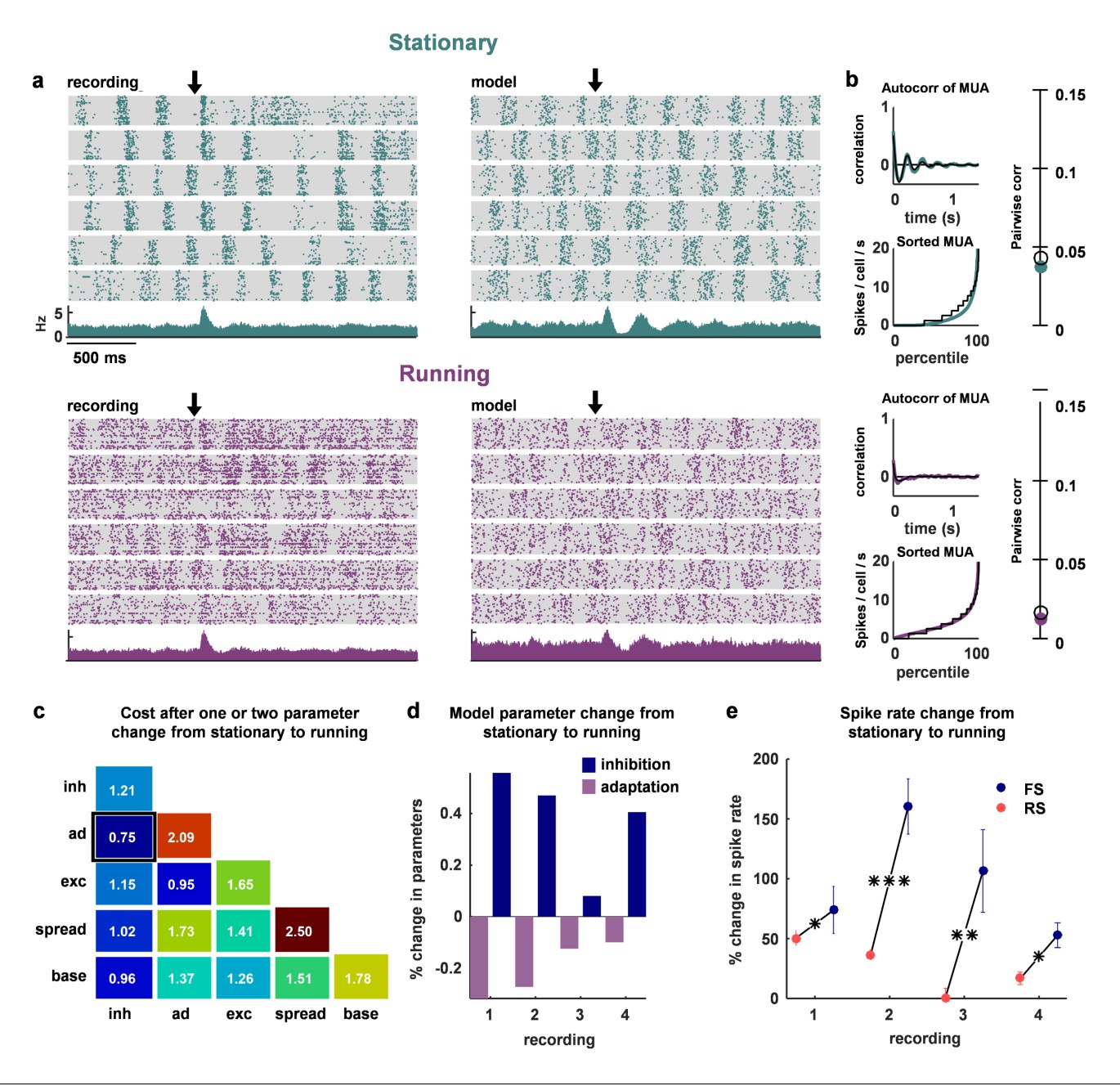

**Figure 7.** The change in dynamics during locomotion is best explained by an increase in inhibition and a reduction in adaptation. (**a**) We recorded populations of neurons in head-fixed mice that were allowed to run on a treadmill. We obtained four separate recordings from two mice, which we divided into running and stationary epochs. The raster plots and PSTHs show evoked responses recorded of one example population when the animal was stationary (top) or running (bottom), along with the activity generated by the network model fit to each set of epochs. The units for the vertical axis on the PSTH are spikes / cell / s. The arrow indicates stimulus onset. (**b**) Model and data summary statistics for stationary (top) and running (bottom) epochs for one example population, plotted as in *Figure 3*. The model fits shown for running epochs were achieved by allowing two parameters (inhibition and adaptation) to change from fits to stationary epochs. (**c**) We fit our network model to activity from stationary epochs and investigated which changes in either one or two parameters best captured the change in dynamics that followed the transition to running. The best achieved cost with changes in each parameter (values along diagonal), or pair of parameters (values off diagonal), is shown (lower is better). (**d**) For the pair of parameters that best described the change in dynamics that followed the transition to running, model inhibition increased and adaptation decreased for each recording. (**e**) The spike rates of both FS and RS neurons were increased by running, but the relative increase was significantly larger for FS neurons in all four recordings (Wilcoxon rank-sum test, $p = 0.043, p < 10^{-5}, p < 10^{-2}, p = 0.037$ respectively). Across all recorded neurons, FS activity increased by 87% and RS activity increased by 28% during running (Wilcoxon rank-sum test, $p < 10^{-6}$).

*Figure 7 continued on next page*

*Figure 7 continued*

The following figure supplement is available for figure 7:

**Figure supplement 1.** Statistics for all fits.

connectivity has been used to create clustered networks (*Doiron et al., 2016*). However, while clustered networks do produce activity with positive correlations between a small fraction of neuron pairs (less than 1 in 1000), the average noise correlations across all pairs are still near zero and, thus, these networks are still unable to generate population-wide fluctuations.

We were able to overcome the limitations of previous models and generate intrinsic large-scale variability that is quantitatively similar to that observed in vivo by using spike-frequency adaptation currents in excitatory neurons, which have been well-documented experimentally (*Nowak et al., 2003*; *Compte et al., 2003*). The population-wide fluctuations generated by the interaction between recurrent excitation and adaptation were a robust feature of the network and persisted in more sophisticated networks that included multiple conductance timescales, many more neurons, spiking inhibitory neurons, structured connectivity, and kurtotic distributions of synaptic efficacies (see *Figure 2—figure supplement 1*).

Although several features of the model network are capable of controlling its intrinsic dynamics, our analysis suggests that differences in feedback inhibition account for the differences in correlations across our in vivo recordings. When we fit the model to each of our individual recordings, we found that noise correlations, as well as stimulus selectivity and decoding accuracy, varied strongly with the strength of inhibition in the network. We also found that the activity of putative inhibitory neurons in our recordings was increased during periods of cortical desynchronization with weak noise correlations. Taken together, these results suggest that the control of correlated variability by inhibition plays a critical role in modulating the impact of intrinsic cortical dynamics on sensory responses.

## Inhibition controls the strength of the large-scale fluctuations that drive noise correlations

Our results are consistent with experiments showing that one global dimension of variability largely explains both the pairwise correlations between neurons (*Okun et al., 2015*) and the time course of population activity (*Ecker et al., 2014*). In our network model, the coordinated, large-scale fluctuations that underlie this global dimension of variability are generated primarily by the interaction between recurrent excitation and adaptation. When inhibition is weak, small deviations from the mean spike rate can be amplified by strong, non-specific, recurrent excitation into population-wide events (up states). These events produce strong adaptation currents in each activated neuron, which, in turn, result in periods of reduced spiking (down states) (*Latham et al., 2000*; *Destexhe, 2009*; *Curto et al., 2009*; *Mochol et al., 2015*). The alternations between up states and down states have an intrinsic periodicity given by the timescale of the adaptation currents, but the chaotic nature of the network adds an apparent randomness to the timing of individual events, thus creating intrinsic temporal variability. Several previous studies (*Tsodyks et al., 1998*; *Loebel et al., 2007*) have modelled alternations between up states and down states using synaptic depression rather than spike-frequency adaptation. However, to our knowledge, there is no experimental evidence for the involvement of synaptic depression in the control of cortical state.

The intrinsic temporal variability in the network imposes a history dependence on evoked responses; because of the build-up of adaptation currents during each spiking event, external inputs arriving shortly after an up state will generally result in many fewer spikes than those arriving during a down state (*Curto et al., 2009*). This history dependence creates a trial-to-trial variability in the total number of stimulus-evoked spikes that is propagated and reinforced across consecutive stimulus presentations to create noise correlations. However, when the strength of the inhibition in the network is increased, the inhibitory feedback is able to suppress some of the amplification by the recurrent excitation, and the transitions between clear up and down states are replaced by weaker fluctuations of spike rate that vary more smoothly over time. If the strength of the inhibition is

increased even further, such that it becomes sufficient to counteract the effects of the recurrent excitation entirely, then the large-scale fluctuations in the network disappear, weakening the history dependence of evoked responses and eliminating noise correlations.

## Strong inhibition sharpens tuning curves and enables accurate decoding by stabilizing network dynamics

Numerous experiments have demonstrated that inhibition can shape the tuning curves of cortical neurons, with stronger inhibition generally resulting in sharper tuning (*Isaacson and Scanziani, 2011*). The mechanisms involved are still a subject of debate, but this sharpening is often thought to result from structured connectivity that produces differences in the tuning of the excitatory and inhibitory synaptic inputs to individual neurons; lateral inhibition, for example, can sharpen tuning when neurons with similar, but not identical, tuning properties inhibit each other. Our results, however, demonstrate that strong inhibition can sharpen tuning in a network without any structured connectivity simply by controlling its dynamics.

In our model, broad tuning curves result from the over-excitability of the network. When inhibition is weak, every external input will eventually excite every neuron in the network because those neurons that receive the input directly will relay indirect excitation to the rest of the network. When inhibition is strong, however, the indirect excitation is largely suppressed, allowing each neuron to respond selectively to only those external inputs that it receives either directly or from one of the few other neurons to which it is strongly coupled. Thus, when inhibition is weak and the network is unstable, different external inputs will trigger similar population-wide events (*Bathellier et al., 2012*), so the selectivity of the network in this regime is weak and its ability to encode differences between sensory stimuli is poor. In contrast, when inhibition is strong and the network is stable, different external inputs will reliably drive different subsets of neurons, and the activity patterns in the network will encode different stimuli with high selectivity and enable accurate decoding.

## Two different dynamical regimes with weak noise correlations

A number of studies have observed that the noise correlations in cortical networks can be extremely weak under certain conditions (*Ecker et al., 2010*; *Renart et al., 2010*; *Hansen et al., 2012*; *Pachitariu et al., 2015*). It was originally suggested that noise correlations were weak because the network was in an asynchronous state in which neurons are continuously depolarized with a resting potential close to the spiking threshold (*Renart et al., 2010*; *van Vreeswijk and Sompolinsky, 1996*). Experimental support for this classical asynchronous state has been provided by intracellular recordings showing that the membrane potential of cortical neurons is increased during locomotion (*McGinley et al., 2015a*) and hyper-arousal (*Constantinople and Bruno, 2011*), resulting in tonic spiking. However, other experiments have shown that the membrane potential of cortical neurons in behaving animals can also be strongly hyperpolarized with clear fluctuations between up and down states (*Sachidhanandam et al., 2013*; *Tan et al., 2014*; *McGinley et al., 2015a*; *Polack et al., 2013*) (for a table listing the species and brain area investigated in each of these studies, see *Supplementary file 2*).

Many forms of arousal tend to reduce the power of these low-frequency fluctuations in membrane potential (*Sachidhanandam et al., 2013*; *Bennett et al., 2013*; *Polack et al., 2013*; *McGinley et al., 2015a*; *Crochet et al., 2011*); however, there is mounting evidence suggesting that different forms of arousal may have distinct effects on neural activity (*McGinley et al., 2015b*). Locomotion in particular tends to depolarize cortical neurons, and in some cases increases tonic spiking (*Niell and Stryker, 2010*). In contrast, task-engagement in stationary animals has been associated with hyperpolarization and suppression of activity (*McGinley et al., 2015a*; *Otazu et al., 2009*; *Buran et al., 2014*) (but not all studies find a decrease in membrane potential during task engagement [*Sachidhanandam et al., 2013*]). The existence of two different dynamical regimes with weak noise correlations was also apparent in our recordings; while some recordings with weak noise correlations resembled the classical asynchronous state with spontaneous activity consisting of strong, tonic spiking (e.g. desynchronized urethane recordings and some awake recordings), other recordings with weak noise correlations exhibited a suppressed state with relatively low spontaneous activity that contained clear, albeit weak, up and down states (e.g. FMM recordings and other awake recordings).

Our model was able to accurately reproduce spontaneous activity patterns and generate evoked responses with weak noise correlations in both of these distinct regimes.

In addition to strong inhibition, the classical asynchronous state with strong, tonic spiking appears to require a combination of weak adaptation and an increase in the number of neurons receiving strong tonic input (see parameter sweeps in *Figure 2c–d* and parameter values for awake mouse V1 recordings in *Figure 3—figure supplement 2*). Since large-scale fluctuations arise from the synchronization of adaptation currents across the population, reducing the strength of adaptation diminishes the fluctuations (*Destexhe, 2009*; *Curto et al., 2009*; *Mochol et al., 2015*). Increasing tonic input also diminishes large-scale fluctuations, but in a different way (*Latham et al., 2000*); when a subset of neurons receive increased tonic input, their adaptation currents may no longer be sufficient to silence them for prolonged periods, and the activity of these neurons during what would otherwise be a down state prevents the entire population from synchronizing. When the network in the asynchronous state is driven by an external input, it responds reliably and selectively to different inputs. Because the fluctuations in the network are suppressed and its overall level of activity remains relatively constant, every input arrives with the network in the same moderately-adapted state, so there is no history dependence to create noise correlations in evoked responses.

Unlike in the classical asynchronous state, networks in the suppressed state have slow fluctuations in their spontaneous activity, and the lack of noise correlations in their evoked responses is due to different mechanisms (see parameter values for gerbil A1 FMM recordings in *Figure 3—figure supplement 2*). The fluctuations in the hyperpolarized network are only suppressed when the network is driven by external input. In our model, this suppression of the correlated variability in evoked responses is caused by the supralinearity of the feedback inhibition (*Rubin et al., 2015*). The level of spontaneous activity driven by the tonic input to each neuron results in feedback inhibition with a relatively low gain, which is insufficient to suppress the fluctuations created by the interaction between recurrent excitation and adaptation. However, when the network is strongly driven by external input, the increased activity results in feedback inhibition with a much higher gain, which stabilizes the network and allows it to respond reliably and selectively to different inputs. This increase in the inhibitory gain of the driven network provides a possible mechanistic explanation for the recent observation that the onset of a stimulus quenches variability (*Churchland et al., 2010*) and switches the cortex from a synchronized to a desynchronized state (*Tan et al., 2014*), as well as for the suppression of responses to high-contrast stimuli in alert animals (*Zhuang et al., 2014*).

## Experimental evidence for inhibitory stabilization of cortical dynamics

The results of several previous experimental studies also support the idea that strong inhibition can stabilize cortical networks and enhance sensory coding. In vitro studies have shown that pharmacologically reducing inhibition increases the strength of the correlations between excitatory neurons in a graded manner (*Sippy and Yuste, 2013*). In vivo whole-cell recordings in awake animals have demonstrated that the stimulus-evoked inhibitory conductance, measured at the soma, is much larger than the corresponding excitatory conductance (*Haider et al., 2013*). This strong inhibition in awake animals quickly shunts the excitatory drive and results in sharper tuning and sparser firing than the balanced excitatory and inhibitory conductances observed under anesthesia (*Wehr and Zador, 2003*; *Haider et al., 2013*). During locomotion, fluctuations in activity are reduced and both inhibitory neurons and excitatory neurons increase their firing, but inhibitory neurons are modulated more strongly in our recordings (*Figure 7*). There is controversy in the literature as to whether somatostatin-positive (SOM+) inhibitory neurons increase their activity during running, but several studies have found an increase in putative parvalbumin-positive (PV+) inhibitory neuron firing during running (*Niell and Stryker, 2010*; *Polack et al., 2013*; *Vinck et al., 2015*; *Pakan et al., 2016*), consistent with our results.

While some of the increased inhibition in awake behaving animals may be due to inputs from other brain areas (*Yu et al., 2015*), the increased activity of local inhibitory interneurons appears to play an important role (*Schneider et al., 2014*; *Kato et al., 2013*; *Kuchibhotla et al., 2016*). However, not all studies have observed increased inhibition in behaving animals (*Zhou et al., 2014*), and the effects of behavioral state on different inhibitory interneuron types are still being investigated (*Gentet et al., 2010*, *2012*; *Polack et al., 2013*). In our model, we ignored the diversity of interneurons in cortex. However, our analyses are generalizable to any interneuron population that may be upregulated during cortical desynchronization. Any interneuron population that exerts a net

inhibitory effect on pyramidal neurons could act to suppress large-scale fluctuations (*Pfeffer et al., 2013*). Determining how each class of inhibitory interneurons contributes to the control of cortical dynamics and modeling those contributions explicitly are important topics for future research.

The effects of local inhibition on sensory coding have also been tested directly using optogenetics. While the exact roles played by different inhibitory neuron types are still under investigation (*Lee et al., 2014*; *Seybold et al., 2015*), the activation of inhibitory interneurons generally results in sharper tuning, weaker correlations, and enhanced behavioral performance (*Wilson et al., 2012*; *Lee et al., 2012*; *Chen et al., 2015*), while suppression of inhibitory interneurons has the opposite effect, decreasing the signal-to-noise ratio and reliability of evoked responses across trials (*Zhu et al., 2015*; *Chen et al., 2015*). These results demonstrate that increased inhibition enhances sensory processing and are consistent with the overall suppression of cortical activity that is often observed during active behaviors (*Otazu et al., 2009*; *Schneider et al., 2014*; *Kuchibhotla et al., 2016*; *Buran et al., 2014*). In fact, one recent study found that the best performance in a detection task was observed on trials in which the pre-stimulus membrane voltage was hyperpolarized and low-frequency fluctuations were absent (*McGinley et al., 2015a*), consistent with a suppressed, inhibition-stabilized network state.

## Acetylcholine and norepinephrine can modulate the inhibitory control of cortical dynamics

Neuromodulators can exert a strong influence on cortical dynamics by regulating the balance of excitation and inhibition in the network. While the exact mechanisms by which neuromodulators control cortical dynamics are not clear, several lines of evidence suggest that neuromodulator release serves to enhance sensory processing by increasing inhibition. Increases in acetylcholine (ACh) and norepinephrine (NE) have been observed during wakefulness and arousal (*Berridge and Waterhouse, 2003*; *Jones, 2008*), and during periods of cortical desynchronization in which slow fluctuations in the LFP are suppressed (*Goard and Dan, 2009*; *Chen et al., 2015*; *Castro-Alamancos and Gulati, 2014*). Stimulation of the basal forebrain has been shown to produce ACh-mediated increases in the activity of FS neurons and decrease the variability of evoked responses in cortex (*Sakata, 2016*; *Castro-Alamancos and Gulati, 2014*; *Goard and Dan, 2009*). In addition, optogenetic activation of cholinergic projections to cortex resulted in increased firing of SOM+ inhibitory neurons and reduced slow fluctuations (*Chen et al., 2015*). The release of NE in cortex through microdialysis had similar effects, increasing fast-spiking activity and reducing spontaneous spike rates (*Castro-Alamancos and Gulati, 2014*), while blocking NE receptors strengthened slow fluctuations in membrane potential (*Constantinople and Bruno, 2011*). More studies are needed to tease apart the effects of different neurotransmitters on pyramidal neurons and interneurons (*Castro-Alamancos and Gulati, 2014*; *Chen et al., 2015*; *Sakata, 2016*), but much of the existing evidence is consistent with our results in suggesting that acetylcholine and norepinephrine can suppress intrinsic fluctuations and enhance sensory processing in cortical networks by increasing inhibition.

## Simulating the neocortical architecture

Recently, there have been major efforts toward constructing neural network simulations of increasingly larger scale (*Izhikevich and Edelman, 2008*) and biological fidelity (*Markram et al., 2015*). There are many biological sources of information that can constrain the parameters of such large-scale simulations, including physiological (*Markram et al., 2015*), anatomical (*Lee et al., 2016*; *Cossell et al., 2015*; *Wertz et al., 2015*) and genetic (*Pfeffer et al., 2013*; *Tasic et al., 2016*). However, while such complex simulations may be able to capture the relevant properties of a circuit and replicate features of its neural activity in detail, they may not necessarily provide direct insight into the general mechanisms that underlie the circuit's function. Thus, a complementary stream of research is needed to seek minimal functional, yet physiologically-based, models that are capable of reproducing relevant phenomena. The model we have investigated here includes only a very restricted set of physiological properties, yet is able to reproduce a wide range of dynamics observed across different species, brain areas, and behavioral states. This simple model provides a compact and intuitive description of the circuit mechanisms that are capable of coordinated dynamics in networks with intrinsic variability. We have already shown that the same mechanisms can also

control the dynamics of more complex functional models, but further work is needed to develop methods to bridge the gap between functional models and large-scale digital reconstructions.

## Materials and methods

All of the recordings analyzed in this study have been described previously, except for the awake V1 data recorded during locomotion. Only a brief summary of the relevant experimental details are provided here. Each recording is considered as a single sample point to which we fit our model. Thus, our sample size is 59. This is justified as sufficient because our samples span multiple brain regions and multiple species, and may be considered as representative activity for a range of different brain states. Due to the sample size, we used the Spearman's (non-parametric) rank correlation in most of our analyses.

### Mouse V1, awake passive

The experimental details for the mouse V1 awake passive recordings have been previously described (*Okun et al., 2015*). The recordings were performed on male and female mice older than 6–7 weeks, of C57BL/6J strain. Mice were on 12 hr non-reversed light-dark regime. The mice were implanted with head plates under anesthesia. After head plate implant each mouse was housed individually. After a few days of recovery the mice were accustomed to having their head fixed while sitting or standing in a custom built tube. On the day of the recording, the mice were briefly anesthetised with isoflurane, and a small craniectomy above V1 was made. Recordings were performed at least 1.5 hr after the animals recovered from the anesthesia. Buzsaki32 or A4×8 silicon probes were used to record the spiking activity of populations of neurons in the infragranular layers of V1.

Visual stimuli were presented on two of the three available LCD monitors, positioned ~25 cm from the animal and covering a field of view of ~120° × 60°, extending in front and to the right of the animal. Visual stimuli consisted of multiple presentations of natural movie video clips. For recordings of spontaneous activity, the monitors showed a uniform grey background.

### Mouse V1, running

Two additional recordings were performed on two female mice, 14 and 20 weeks old. These mice expressed ChR2 in PV+ neurons (Pvalbtm1(cre)Arbr driver crossed with Ai32 reporter). Mice were on 12 hr non-reversed light-dark regime. The mice were implanted with head plates under anesthesia. After head plate implant each mouse was housed individually. After a few days of recovery the mice were accustomed to having their head fixed while standing or running on a styrofoam treadmill. On the day of the recording, the mice were briefly anesthetized with isoflurane and a small craniectomy was made above V1. Recordings were performed at least 1.5 hr after recovery from the anesthesia. Mice were head-fixed above the treadmill and allowed to run at will while multi-neuron recordings were made across all layers using probes that were inserted roughly perpendicular to the cortical surface. Raw voltage signals were referenced against an Ag/AgCl wire in a saline bath above the craniectomy, amplified with analog Intan amplifiers, and digitized at 25 kHz with a WHISPER acquisition system. Visual stimuli were presented on three LCD monitors, positioned as three sides of a square, 20 cm from the animal and covering a field of view of approximately 270 x 70, centered on the direction of the mouses nose (*Okun et al., 2015*). Visual stimuli consisted of drifting gratings of different sizes, approximately centered on the receptive field location of the recorded neurons. Stimuli were either 1 or 2 s long, and in the periods between stimuli (durations of 0.4–1 s), the monitors showed a uniform grey background.

### Rat A1

The experimental procedures for the rat A1 recordings have been previously described (*Luczak et al., 2009*). Briefly, head posts were implanted on the skull of male Sprague Dawley rats (300–500 g, normal light cycle, regular housing conditions) under ketamine-xylazine anesthesia, and a hole was drilled above the auditory cortex and covered with wax and dental acrylic. After recovery, each animal was trained for 6–8 d to remain motionless in the restraining apparatus for increasing periods (target, 1–2 hr). On the day of the recording, each animal was briefly anesthetized with isoflurane and the dura resected; after a 1 hr recovery period, recording began. The recordings were made from infragranular layers of auditory cortex with 32-channel silicon multi-tetrode arrays.

Sounds were delivered through a free-field speaker. As stimuli we used pure tones (3, 7, 12, 20, or 30 kHz at 60 dB). Each stimulus had duration of 1 s followed by 1 s of silence. All procedures conformed to the National Institutes of Health Guide for the Care and Use of Laboratory Animals.

## Gerbil A1

The gerbil A1 recordings have been described in detail previously (*Pachitariu et al., 2015*). Briefly, adult male gerbils (70–90 g, P60-120, normal light-dark cycle, group housed) were anesthetized with one of three different anesthetics: ketamine/xylazine (KX), fentanyl/medetomidine/midazolam (FMM), or urethane. A small metal rod was mounted on the skull and used to secure the head of the animal in a stereotaxic device in a sound-attenuated chamber. A craniotomy was made over the primary auditory cortex, an incision was made in the dura mater, and a 32-channel silicon multi-tetrode array was inserted into the brain. Only recordings from A1 were analyzed. Recordings were made between 1 and 1.5 mm from the cortical surface (most likely in layer V). All gerbils recorded were used in this study, except for one gerbil under FMM which exhibited little to no neural activity during the recording period.

Sounds were delivered to speakers coupled to tubes inserted into both ear canals for diotic sound presentation along with microphones for calibration. Repeated presentations of a 2.5 s segment of human speech were presented at a peak intensity of 75 dB SPL. For analyses of responses to different speech tokens, seven 0.25 s segments were extracted from the responses to each 2.5 s segment.

## Gerbil IC

The gerbil IC recordings have been described in detail previously (*Garcia-Lazaro et al., 2013*). Recordings were made under ketamine/xylazine anesthesia using a multi-tetrode array placed in the low-frequency laminae of the central nucleus of the IC. Experimental details were otherwise identical to those for gerbil A1. In addition to the human speech presented during the A1 recordings, tones with a duration of 75 ms and frequencies between 256 Hz and 8192 Hz were presented at intensities between 55 and 85 dB SPL with a 75 ms pause between each presentation.

All relevant data are available from the authors upon request.

## Spike sorting and filtering

Details of spike sorting for most recordings have been described in detail before in the respective original publications. Briefly, recordings from mouse V1 (awake passive) and rat A1 were spike sorted with KlustaKwik and further manually inspected in KlustaViewa. Recordings from gerbil A1 or IC were spike sorted with a custom-modified version of KlustaKwik. The unpublished recordings from mouse V1 (awake running) were spike sorted with Kilosort using the default settings (*Pachitariu et al., 2016*) and inspected in Phy (*Rossant et al., 2016*). Only units with spike rates above 0.1 Hz were considered in the analysis. The spike waveforms considered in the FS/RS classification for all recordings were obtained from the Kilosort templates, which correspond to the mean spike shapes.

## Spiking network model

We developed a network model using conductance-based quadratic integrate and fire neurons. There are three currents in the model: an excitatory, an inhibitory and an adaptation current. The subthreshold membrane potential for a single neuron $i$ obeys the equation

$$\tau_m \frac{dV_i}{dt} = -(V_i - E_L)(V_i - V_{th}) - g_{E_i}(V_i - E_E) - g_{I_i}(V_i - E_I) - g_{A_i}(V_i - E_A).$$

When $V > V_{th}$, a spike is recorded in the neuron and the neuron's voltage is reset to $V_{reset} = 0.9 V_{th}$. For simplicity, we set $V_{th} = 1$ and the leak voltage $E_L = 0$. The excitatory voltage $E_E = 2V_{th}$ and $E_I = E_A = -0.5V_{th}$. Each of the conductances has a representative differential equation. The excitatory conductance obeys

$$\tau_E \frac{d\mathbf{g}_E}{dt} = -\mathbf{g}_E + J\mathbf{s} + \mathbf{b}.$$

where $J$ is the matrix of excitatory connectivity and $\mathbf{b}$ is the vector of tonic inputs to the neurons. The matrix of connectivity is random with a probability of 5% for the network of 512 neurons and their connectivities are randomly chosen from a uniform distribution between 0 and $w_E$. The tonic inputs $\mathbf{b}$ have a minimum value $b_0$, which we call the tonic input baseline added to a random draw from an exponential distribution with mean $b_1$, which we call the tonic input spread, such that for neuron $i$, $\mathbf{b}(i) = b_0 + \mathrm{exprnd}(b_1)$. The inhibitory conductance obeys

$$\tau_I \frac{\mathrm{d}g_I}{\mathrm{d}t} = -g_I + w_I(\exp(c\sum \mathbf{s}) - 1).$$

where $c$ controls the gain of the inhibitory conductance. The inhibitory conductance is global, i.e. each neuron receives the same inhibitory feedback, and it obeys an exponential supralinearity (*Rubin et al., 2015*).

The adaptation conductance obeys

$$\tau_A \frac{d\mathbf{g}_A}{\mathrm{d}t} = -\mathbf{g}_A + w_A\mathbf{s}.$$

The simulations are numerically computed using Euler's method with a time-step of 0.75 ms (this was the lock-out window used for spike-sorting the in vivo recordings and allowed for fast simulations). To avoid numerical instabilities at low voltages, we rectified the voltages at the activation potential of the inhibitory conductance. Each parameter set was simulated for 900 s. The timescales are set to $\tau_m = 20$ ms, $\tau_E = 5.10$ ms, $\tau_I = 3.75$ ms, $\tau_A = 375$ ms, and the inhibitory non-linearity controlled by $c = 0.25$. The remaining five parameters ($w_I$, $w_A$, $w_E$, $b_1$, and $b_0$) were fit to the spontaneous activity from multi-neuron recordings using the techniques described below. Their ranges were (0.01–0.4), (0.4–1.45), (2.50–5.00), (0.005–0.10), and (0.0001–0.05) respectively.

To illustrate the ability of the network to generate activity patterns with macroscopic variability, we simulated spontaneous activity with a parameter set that produces up and down state dynamics. *Figure 2a* shows the membrane potential of a single neuron in this simulation and its conductances at each time step. *Figure 2b* shows the model run twice with the same set of initial conditions and parameters, but with an additional single spike inserted into the network on the second run (circled in green).

This code will be made available for use after publication.

## Parameter sweep analysis

*Figure 2c and d* summarize the effects of changing each parameter on the structure of the spontaneous activity patterns generated by the model. We held the values for all but one parameter fixed and swept the other parameter across a wide range of values. The fixed parameter values were set to approximately the median values obtained from fits to all in vivo recordings. *Figure 5b,d and f* and *Figure 4—figure supplement 1* show the results of similar parameter sweep analyses for stimulus-driven activity with the external input to the network derived from IC activity as described below. For these analyses, the values of the parameters that were not swept were fixed at those fit to each individual recording.

## GPU implementation

We accelerated the network simulations by programming them on graphics processing units (GPUs) such that we were able to run them at 650x real time with 15 networks running concurrently on the same GPU. We were thus able to simulate ≈10,000 s of simulation time in 1 s of real time. To achieve this acceleration, we took advantage of the large memory bandwidth of the GPUs. For networks of 512 neurons, the state of the network (spikes, conductances and membrane potentials) can be stored in the very fast shared memory available on each multiprocessor inside a GPU. A separate network was simulated on each of the 8 or 15 multiprocesssors available (video cards were GTX 690 or Titan Black). Low-level CUDA code was interfaced with Matlab via mex routines.

## Summary statistics

Several statistics of spikes were used to summarize the activity patterns observed in the in vivo recordings and in the network simulations. Because there were on the order of 50 neurons in each recording, all of the statistics below were influenced by small sample effects. To replicate this bias in

the analysis of network simulations, we subsampled 50 neurons from the network randomly and computed the same statistics we computed from the in vivo recordings.

The noise correlations between each pair of neurons in each recording were measured from responses to speech. The response of each neuron to each trial was represented as a binary vector with 15 ms time bins. The total correlation for each pair of neurons was obtained by computing the correlation coefficient between the actual responses. The signal correlation was computed after shuffling the order of repeated trials for each time bin. The noise correlation was obtained by subtracting the signal correlation from the total correlation.

The multi-unit activity (MUA) was computed as the sum of spikes in all neurons in bins of 15 ms. The autocorrelation function of the MUA at time-lag $\tau$ was computed from the formula

$$\mathrm{ACF}(\tau) = \frac{1}{N_{samples}} \sum \mathrm{MUA}(t) * \mathrm{MUA}(t+\tau)$$

In the awake recordings (mouse V1 passive and running, rat A1) we observed slow-timescale fluctuations on the order of tens of seconds, which significantly affected the autocorrelation function of the MUA at lags <1 s. We chose to ignore these fluctuations during model fitting by high-pass filtering the MUA at 1 Hz before computing the autocorrelation function.

To measure the autocorrelation timescale, we fit one side of the ACF with a parametric function

$$\mathrm{ACF}(\tau) \sim a \exp(-\tau/T) \cdot \cos(\tau/(2\pi t_{\mathrm{period}}))$$

where $a$ is an overall amplitude, $T$ a decay timescale and $t_{\mathrm{period}}$ is the oscillation period of the autocorrelation function. There was not always a significant oscillatory component in the ACF, but the timescale of decay accurately captured the duration over which the MUA was significantly correlated.

## Parameter searches

To find the best fit parameters for each individual recording, we tried to find the set of model parameters for which the in vivo activity and the network simulations had the same statistics. We measured goodness of fit for each of the three statistics: pairwise correlations, the MUA distribution, the MUA ACF. Each statistic was normalized appropriately to order 1, and the three numbers obtained were averaged to obtain an overall goodness of fit.

The distance measure $D_c$ between the mean correlations $c_\theta$ obtained from a set of parameters $\theta$ and the mean correlations $c_n$ in recording $n$ was simply the squared error $D_c(c_n, c_\theta) = (c_n - c_\theta)^2$. This was normalized by the variance of the mean correlations across recordings to obtain the normalized correlation cost $\mathrm{Cost}_c$, where $\langle x_n \rangle_n$ is used to denote the average of a variable $x$ over recordings indexed by $n$.

$$\mathrm{Cost}_c = \frac{D_c(c_n, c_\theta)}{\langle D_c(c_n, \langle c_n \rangle) \rangle}$$

The distance measure $D_m$ for the MUA distribution was the squared difference summed over the order rank bins $k$ of the distribution $D_m(\mathrm{MUA}_n, \mathrm{MUA}_\theta) = \sum_k (\mathrm{MUA}_n(k) - \mathrm{MUA}_\theta(k))^2$. This was normalized by the distance between the data MUA and the mean data MUA. In other words, the cost measures how much closer the simulation is to the data distribution than the average of all data distributions.

$$\mathrm{Cost}_m = \frac{D_m(\mathrm{MUA}_n, \mathrm{MUA}_\theta)}{D_m(\mathrm{MUA}_n, \langle \mathrm{MUA}_n \rangle)}$$

Finally, the distance measure $D_a$ for the autocorrelation function of the MUA was the squared difference summed over time lag bins $t$ of the distribution $D_a(\mathrm{ACF}_n, \mathrm{ACF}_\theta) = \sum_t (\mathrm{ACF}_n(t) - \mathrm{ACF}_\theta(t))^2$.

This was normalized by the distance between the data ACF and the mean data ACF.

$$\mathrm{Cost}_a = \frac{D_a(\mathrm{ACF}_n, \mathrm{ACF}_\theta)}{D_a(\mathrm{ACF}_n, \langle \mathrm{ACF}_n \rangle)}$$

The total cost of parameters $\theta$ on recording $n$ is therefore $\mathrm{Cost}(n, \theta) = \mathrm{Cost}_c + \mathrm{Cost}_m + \mathrm{Cost}_a$.

Approximately one million networks were simulated on a grid of parameters for 900 s each of spontaneous activity, and their summary statistics ($c_\theta$, $\mathrm{MUA}_\theta$ and $\mathrm{ACF}_\theta$) were retained. The Cost was smoothed for each recording by averaging with the nearest 10 other simulations on the grid. This ensured that some of the sampling noise was removed and parameters were estimated more robustly. The best fit set of parameters was chosen as the minimizer of this smoothed cost function, on a recording by recording basis.

### Evaluation of the goodness-of-fit of the model

We computed the upper limit for the explained variance of the model based on the recordings. We split each neural recording into two halves (interleaved segments of 4 s each) and computed the amount of variance in statistics from one half of the recording that is explained by the other half of the recording. We compared this to the amount of variance in the statistics of the full recording that was explained by the model.

### Alternative Gibbs sampling parameter optimization

We also demonstrate an alternative approach to finding the best fitting parameters through a sampling-based optimization procedure (*Figure 3—figure supplement 1*). This reduces the necessary number of simulations from 1 million to 100,000. Future work might in principle devise even faster optimizations, thus allowing analysis on a bigger scale than presented here. Briefly, the sampling-based optimization is based on defining the energy landscape as the negative of the cost function, and thus defining a probability distribution over parameters $\mathrm{P}(\theta) = \exp(-\mathrm{Cost}(\theta)/T)$, where $T$ is the temperature. We use a proposal distribution that always proposes neighbors of the current sample on the grid on which we did the full parameter sweeps, and accept the proposals according to the balance equations of Markov Chain Monte Carlo sampling (MCMC):

$$\mathrm{Prob(accept)} = \frac{\mathrm{P}(\theta_{\mathrm{new}})}{\mathrm{P}(\theta_{\mathrm{new}}) + \mathrm{P}(\theta_{\mathrm{old}})}$$

$$= \frac{1}{1 + \exp(-(\mathrm{Cost}(\theta_{\mathrm{new}}) - \mathrm{Cost}(\theta_{\mathrm{old}}))/\mathrm{T})}$$

To avoid the MCMC chains getting stuck into low probability parts of the energy landscape, we restart the chain every 50 samples from the pool of already-sampled points, chosen with probability proportional to its $\mathrm{P}(\theta)$. Furthermore, we allow the chain used to optimize the model parameters for one recording to use information from the chains used for the other recordings by pooling together the already-sampled points from all datasets and restarting chains based on all these points.

### NMDA and GABA$_B$ conductance network

We added long timescale excitatory and inhibitory conductances to the model and simulated the model at multiple levels of inhibitory feedback strength. The strength of the NMDA conductance was 4% the strength of the AMPA conductance and $\tau_{NMDA} = 100$ ms (thus the integrated current was approximately the same as the AMPA integrated current injection). The strength of the GABA$_B$ conductance was 2% the strength of the GABA conductance and it had the same timescale as NMDA. The parameter set used for *Figure 2—figure supplement 1a* was $\theta$ = (0.51, $w_I$, 2.6, 0.008, 0.037), where $w_I$ ranged from 0.02 to 0.25.

### Clustered neuronal network with intrinsic variability and spiking inhibitory neurons

We also simulated a clustered architecture with variability and adaptation currents. This model consisted of 144 clusters, each with 32 neurons, eight of which were inhibitory neurons and 24 of which were excitatory neurons. The probability of within cluster excitatory-excitatory (E-E) connectivity was 0.3, and within cluster inhibitory-excitatory (I-E) and excitatory-inhibitory (E-I) were 0.15 and 0.1 respectively. The probability of out of cluster E-E, I-E, and E-I connectivity were 0.012, 0.03, and 0.01 respectively. The inhibitory-inhibitory connectivity was unclustered. The probability of connection was 0.01 and its strength was 0.17. The average connection strengths for E-E and I-E were 0.024 and 0.016 respectively. The E-I strength in *Figure 2—figure supplement 1b* ranged from

0.025 to 0.057. The adaptation current had strength 0.45 and $\tau_A = 220$ ms. The membrane timescale for excitatory and inhibitory neurons were 25 ms and 5 ms respectively, and $\tau_E = 6$ ms and $\tau_I = 3$ ms.

## Stimulus-driven activity

Once the simulated networks were fit to the spontaneous neuronal activity, we drove them with an external input to study their evoked responses. The stimulus was either human speech (as presented during our gerbil A1 recordings) or pure tones. The external input to the network was constructed using recordings from 563 neurons from the inferior colliculus (IC). For all recordings in the IC the mean pairwise noise correlations were near-zero and the Fano Factors of individual neurons were close to 1 (*Garcia-Lazaro et al., 2013*), suggesting that responses of IC neurons on a trial-by-trial basis are fully determined by the stimulus alone, up to Poisson-like variability. Thus, we averaged the responses of IC neurons over trials and drove the cortical network with this trial-averaged IC activity. We binned IC neurons by their preferred frequency in response to pure tones, and drove each model cortical neuron with a randomly chosen subset of 10 neurons from the same preferred-frequency bin. We rescaled the IC activity so that the input to the network had a mean value of 0.06 and a maximum value of 0.32, which was three times greater than the average tonic input.

We kept the model parameters fixed at the values fit to spontaneous activity and drove the network with 330 repeated presentations of the stimulus. We then calculated the statistics of the evoked activity. Noise correlations were measured in 15 ms bins as the residual correlations left after subtracting the mean response of each neuron to the stimulus across trials:

$$c_{ij} = \frac{1}{N_{samples}} \sum_t (s_i(t) - <s_i(t)>)(s_j(t) - <s_j(t)>)$$

where $s_i(t)$ is the summed spikes of neuron $i$ in a 15 ms bin and $<s_i(t)>$ is the mean response of neuron $i$ to the stimulus. The noise correlation value given for each recording is the mean of $c_{ij}$.

## Tuning width

To determine tuning width to sound frequency, we used responses of IC neurons to single tones as inputs to the model network. The connections from IC to the network were the same as described in the previous section. Because the connectivity was tonotopic and IC responses are strongly frequency tuned, the neurons in the model network inherited the frequency tuning. We did not model the degree of tonotopic fan-out of connections from IC to cortex and, as a result, the tuning curves of the model neurons were narrow relative to those observed in cortical recordings (*Pachitariu et al., 2015*). We chose the full width of the tuning curve at half-max as a standard measure of tuning width.

## Decoding tasks

We computed decoding error for a classification task in which the single-trial activity of all model neurons was used to infer which of seven different speech tokens was presented. The classifier was built on training data using a linear discriminant formulation in which the Gaussian noise term was replaced by Poisson likelihoods. Specifically, the activity of a neuron for each 15 ms bin during the response to each token was fit as a Poisson distribution with the empirically-observed mean. To decode the response to a test trial, the likelihood of each candidate token was computed and the token with the highest likelihood was assigned as the decoded class. This classifier was chosen because it is very fast and can be used to model Poisson-like variables, but we also verified that it produced decoding performance as good as or better than classical high-performance classifiers like support vector machines.

## Classifying FS and RS neurons

We classified fast-spiking and regular-spiking neurons based on their raw, unfiltered, spike shape (*Okun et al., 2015*). We determined the trough-to-peak time of the mean spike waveform after smoothing with a gaussian kernel of $\sigma$=0.5 samples. The distribution of the trough-to-peak time $\tau$ was clearly bimodal in all types of recordings. Following (*Okun et al., 2015*) we classified FS neurons in the awake data with $\tau$<0.6 ms and RS neurons with $\tau$>0.8 ms. The distributions of $\tau$ in the

anesthetized data, although bimodal, did not have a clear separation point, so we conservatively required τ<0.4 ms to classify an FS cell in these recordings and τ>0.65 ms to classify RS neurons (see *Figure 6—figure supplement 1*). The rest of the neurons were not considered for the plots in *Figures 6* and *7* and are shown in gray on the histogram in *Figure 6—figure supplement 1*.

Although one recent study has raised doubts on the accuracy of spike-width based classification (*Moore and Wehr, 2013*), a large number of other studies have shown 90–100% classification accuracy of FS neurons as PV+ interneurons (*Nowak et al., 2003*; *Kawaguchi and Kubota, 1997*; *Barthó et al., 2004*; *Cho et al., 2010*; *Madisen et al., 2012*; *Stark et al., 2013*; *Cardin et al., 2009*; *Cohen and Mizrahi, 2015*). Even (*Moore and Wehr, 2013*) show that the classification is near-perfect using other features of the spike waveform; their finding that spike-width based classification was not accurate may be due to the filtering that they performed during pre-processing. In our recordings, the distributions of the trough-to-peak duration of the raw waveform are highly bimodal in all cases (see *Figure 6—figure supplement 1*), unlike the distributions shown in (*Moore and Wehr, 2013*).

## Local field potential

The low-frequency potential (LFP) was computed by low-pass filtering the raw signal with a cutoff of 300 Hz. Spectrograms with adaptive time-frequency resolution were obtained by filtering the LFP with Hamming-windowed sine and cosine waves and the spectral power was estimated as the sum of their squared amplitudes. The length of the Hamming window was designed to include two full periods of the sine and cosine function at the respective frequency, except for frequencies of 1 Hz and above 30 Hz, where the window length was clipped to a single period of the sine function at 1 Hz and two periods of the sine function at 30 Hz respectively. The synchrony level was measured as the log of the ratio of the low to high frequency power (respective bands: 3–10 Hz and 11–96 Hz, excluding 45–55 Hz to avoid the line noise). We did not observe significant gamma power peaks except for the line noise, in either the awake or anesthetized recordings.

## Dividing trials by synchrony

For the recordings from awake restrained mice, we computed a synchrony value for each trial in the 500 ms window following stimulus onset. The distribution of synchrony values was not clearly bimodal, but varied across a continuum of relatively synchronized and desynchronized states. To examine the effect of synchrony on noise correlations, we sorted all trials by their synchrony value, classified the 50% of trials with the lowest values as desynchronized and the 50% of trials with the highest values as synchronized, and computed the noise correlations for each set of trials for each recording. To examine the effect of synchrony on FS and RS activity, we pooled all trials from all recordings, divided them into quintiles by their synchrony value, and computed the average spike rates of FS and RS neurons for each set of trials. For *Figure 6*, noise correlations were computed aligned to the stimulus onsets in windows of 500 ms, to match the window used for measuring FS and RS activity as well as LFP power.

For urethane recordings, we computed the level of synchrony of the LFP (ratio of low frequency 1–10 Hz activity to high frequency 11–100 Hz) in sliding 10 s windows. The recordings were split into high and low synchrony based on the median level of synchrony, and 20 s around each transition point were discarded. We treated urethane recordings in synchronized and desynchronized states as separate recordings for the purposes of model fitting.

## Dividing trials by behavioral state

We median-filtered the raw running speed of the treadmill with a window of 0.5 s. In order to discard extremely small speeds that may be noise, we discarded all points less than one hundredth of the standard deviation of the running speed. Using the processed running speed, we divided the data into periods of stationary and running behavior. We found periods of at least five seconds in which all bins were either zero or non-zero. We then excluded the first and last second of each of these segments from our computations, considering them to be periods of transition between stationary and running. We took all of the spiking in these segments and binned it into 15 ms bins for all further computations.

## Modeling the transition from stationary to running

In order to fit the intrinsic variability in the population responses in the recordings that were made on the treadmill, we first removed the evoked responses from each recording. We computed the mean evoked response to each stimulus (10 stimuli total) for each neuron by computing the mean response across all trials and then subtracting the spontaneous spike rate. We then subtracted this mean evoked response from each neurons response on each trial. Because the spike rates of neurons varied from 0.1 Hz (our cutoff for inclusion) to 150 Hz, we divided each neuron's binned spike rate by its average spike rate and then multiplied by the overall mean spike rate of the population. We then computed each of our statistics on these normalized population activity patterns. We fit the models to the statistics of the stationary periods in each of the recordings, and then changed one or two parameters of the stationary fits in order to best fit the running periods. We simulated evoked activity by driving the model with the mean evoked responses described above, scaled so that the overall spike rate of the model responses matched the overall spike rate in each recording.

# Additional information

### Funding

| Funder | Grant reference number | Author |
|---|---|---|
| Gatsby Charitable Foundation | | Carsen Stringer<br>Marius Pachitariu<br>Maneesh Sahani |
| Wellcome Trust | 095668 | Marius Pachitariu<br>Nicholas A Steinmetz<br>Kenneth D Harris |
| Wellcome Trust | 095669 | Marius Pachitariu<br>Nicholas A Steinmetz<br>Kenneth D Harris |
| Wellcome Trust | 100154 | Marius Pachitariu<br>Nicholas A Steinmetz<br>Kenneth D Harris |
| Simons Foundation | 325512 | Marius Pachitariu<br>Nicholas A Steinmetz<br>Kenneth D Harris |
| Hungarian Brain Research Program | KTIA NAP 13-2-2014-0016 | Peter Bartho |
| Simons Foundation | SCGB 323228 | Maneesh Sahani |
| Wellcome Trust | Research Fellowship WT086697MA | Nicholas A Lesica |
| Wellcome Trust | 200942/Z/16/Z | Nicholas A Lesica |

The funders had no role in study design, data collection and interpretation, or the decision to submit the work for publication.

### Author contributions

CS, MP, KDH, MS, Conception and design, Analysis and interpretation of data, Drafting or revising the article; NAS, Conception and design, Acquisition of data; MO, PB, Conception and design, Acquisition of data, Drafting or revising the article; NAL, Conception and design, Acquisition of data, Analysis and interpretation of data, Drafting or revising the article

### Author ORCIDs

Carsen Stringer, http://orcid.org/0000-0002-9229-4100
Maneesh Sahani, http://orcid.org/0000-0001-5560-3341
Nicholas A Lesica, http://orcid.org/0000-0001-5238-4462

## Ethics

Animal experimentation: This study involved analysis of previously published and new data. All new experiments were performed in accordance with the UK Animal (Scientific Procedures) Act under project license 70/8021. All surgeries were performed under anesthesia, and every effort was made to minimize suffering.

# Additional files

### Supplementary files

• Supplementary file 1. Metadata for all recordings. The table shows the species, brain region, and state for each of our recordings, as well information regarding the stimuli presented and the number of neurons recorded.

• Supplementary file 2. Species and brain area for studies of cortical dynamics cited in the Discussion. The table shows the species and brain region for the studies cited in the sections of the Discussion related to cortical dynamics.

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
