## [Decision Letter]

Thank you for submitting your article "Inhibitory control of correlated intrinsic variability in cortical networks" for consideration by *eLife*. Your article has been favorably evaluated by Eve Marder (Senior Editor) and three reviewers, one of whom is a member of our Board of Reviewing Editors. The reviewers have opted to remain anonymous.

The reviewers have discussed the reviews with one another and the Reviewing Editor has drafted this decision to help you prepare a revised submission.

Summary:

In this manuscript, Stringer et al. and a team of theorists examine cortical activity recordings from several species (gerbil, mouse, and rat), several cortical areas (auditory and visual), and several anesthetic/behavioural states, in which correlations (particularly those driven by Up/Down state dynamics) are more or less prominent. They construct a spiking network model that can generate Up/Down dynamics internally (the mechanism is that neurons' afterhyperpolarizations chaotically synchronize) by systematically varying a handful of key parameters of the model. They predict that the main feature determining whether Up/Down dynamics are prominent is the strength of feedback inhibition. They then test this prediction by comparing activity of fast-spiking versus broad-spiking units in the original data, and they see that FS units are indeed more responsive during desynchronized trials or under fentanyl, versus on synchronized trials or under ketamine.

The comprehensive assessment of the relationship between inhibition, strength of correlations, and synchronization in cortical states is of fundamental importance, and the attempt to perform such assessment rigorously and quantitatively is a significant and novel step. However, the results paint an oversimplified picture that glosses over many well-established experimental findings that complicate the interpretation and at least sometimes contradict the results. This is partly because the model makes some substantial simplifications (e.g., a single source of inhibition), partly because the literature itself is complex and contradictory, and partly because the data that agree with the model are overemphasized.

As it is, the manuscript may be significant for other modeling groups, but the impact for experimentalists will be very minor due to the difficulty of interpretation. The consensus of the reviewers is that addressing/discussing the limitations of the model (as suggested below) in a substantial way would enormously enhance the impact of this work.

Essential revisions:

1) Both the chosen statistics and the qualitative comparisons between data and model (e.g. Figure 3) indicate very good agreement. However, one still wonders, how good can the fits be? And, what is the limiting factor, the dynamical repertoire of the model or the amount of data? These are important questions for determining not only the applicability of the model, but also what it would take to improve upon the current results.

Here is one way such a comparison could be made. The idea is to generate an estimate of the lowest possible error (or cost) given the data, and compare it to the actual error between model and data (for any given recording). To get such limit, divide the recorded data set into two equal parts, A and B (say, the first and second halves of the total recording time), and compare them using the same exact statistics and cost functions used to compare the model and the full data set, autocorrelation function, correlations, etc. If the total error between parts A and B is, say, 0.10, and that between model and data is, say, 0.20 (i.e., 100% larger), then we know that the model probably still has room to improve, i.e., one could fiddle with other parameters or make it more complex in order to generate a better match to the data. In contrast, if the error between model and data is, say, 0.11 (just 10% higher than that between parts A and B), it would suggest that the model is essentially matching the data as well as that can be done given the amount at hand, and the chosen statistics.

This, or something like this, would provide a more intuitive sense for how good the model is in relation to how good any model *can* be under the circumstances. It would indicate how drastic are the simplifications of the model. For instance, if the error between model and data is far higher than the bound found by randomization, it would suggest that less drastic simplifications would be able to capture more variance in the data.

2) The model would be much more useful if the results were more interpretable. Perhaps the authors could unpack one of the areas of simplification and explore how it affects the results. Here are two possibilities.

One, the findings would be more interpretable if they relied more on naturally occurring cortical states. Not only is the exact pharmacological action of the anesthetics still unclear, there is evidence that some drugs like ketamine act differentially on excitatory and inhibitory neurons, making interpretation of the model difficult. The model would provide a more powerful framework if the authors included recordings where a set of stable, identified neurons was followed across multiple quantified waking states (which could be distinguished by arousal state or EEG/LFP data), or from wakefulness to sleep.

Two, the authors assume that the activity of FS cells represents the state of inhibition in the cortex, and therefore conclude that the level of inhibition is low under some circumstances and higher in others. However, there are several inhibitory-inhibitory connections between different interneuron cell types (PV, SST, VIP etc.). The overall level of inhibition at any given moment may thus remain stable even as the contributing population of interneurons changes. Using a single feedback inhibitory mechanism may not be the most appropriate way to model inhibitory dynamics in the cortex.

For instance, one potential pitfall of incorporating a single feedback inhibition mechanism is that tuning might be sharpened simply by an 'iceberg' effect, where nonspecific increases in inhibition simply eliminate small responses to input. This was noted by Lee et al. 2013 and El-Boustani et al. 2014, following the Lee et al. 2012 paper cited by the authors. The authors should address this issue in interpreting the results from Figure 5.

3) There are several major inconsistencies between the experimental literature and model assumptions that should be discussed.

Regarding the claim that cortical activity is generally suppressed during active behaviors, all the papers cited in support of this claim are auditory (subsection “Neuromodulators and inhibitory control of cortical dynamics”). However, the exact opposite result is found in the visual system, in which cortical activity, RS response rate, SNR, and so on are all increased during locomotion. (There are numerous papers on this; Niell/Stryker 2010 is a particularly famous example.) This work doesn't fit with their theory at all! Rather than gloss over this issue, the authors should describe the relevant data both for and against their theory, and explain why their theory is still valuable even if not all its predictions are borne out.

Another example is their suggestion that many FS cells do not respond to stimuli under ketamine anesthesia, and that this silencing is responsible for the tendency of cortical networks to generate Up/Down dynamics under ketamine. The issue of whether FS cells are responsive under ketamine was examined by Moore/Wehr 2013 (who identified PV cells using optotagging in mouse auditory cortex under ketamine anesthesia, and showed that most tagged cells had high spontaneous and evoked firing rates), as well as Cohen/Mizrahi 2013 (who randomly patched neurons in ketamine-anesthetized rat barrel cortex, identified the FS neurons by intracellular current injection, and showed that all FS cells did respond either to ipsi- or contra-lateral whisker stimulation). How is the model consistent with these data – or if it's not, can the authors discuss why the model is nonetheless worth considering?

Regarding waveform classification (subsection “Classifying FS and RS neurons”): There is a genuine debate in the literature regarding the relationship between cell type and extracellularly recorded spike waveform, but one wouldn't get that impression from reading this paragraph: first, because only one paper is cited on the "exercise caution" side, and second, because no papers at all are cited to support the preferred estimate of 90-100% correct classification. It would be much better to cite papers on both sides of the debate, since this classification is so central to the analysis.

Regarding whether the FS/RS distinction "is still effective for approximating the overall levels of inhibitory and excitatory activity in a population" (subsection “Activity of fast-spiking (FS) neurons is increased during periods of cortical desynchronization with weak noise correlations”): It is not necessarily true that activity of all types of inhibitory cell will be correlated with one another, particularly since inhibitory cells can inhibit one another in targeted, type-specific ways (e.g. Pfeffer/Scanziani 2013), which will tend to produce anticorrelations between activity in the different classes of interneurons. Fanselow/Connors 2008 showed that the states in which somatostatin cells were active tended to be those in which parvalbumin cells were least active, and in fact Tan/Agmon 2008 suggest that non-FS inhibitory cells (somatostatin cells) are the ones most specialized to provide the kind of stabilizing influence the authors' models require.

None of this implies that there is something wrong with the theory itself. But half the paper is about whether the experiments back up the theory, so caution should be exercised when interpreting the experimental data. As it is, it would appear that the literature provides straightforward, uncontroversial support of the theory, which is not the case.

4) Subsection “Activity of fast-spiking (FS) neurons is increased during periods of cortical desynchronization with weak noise correlations”, last paragraph/Figure 7 shows differences in the total numbers of thin versus broad spikes under fentanyl versus ketamine anesthesia. Many factors can influence how many units/clusters are recovered from a recording and how well spikes are assigned to their underlying sources, including physical/pulsatile instability of the brain caused by breathing and heart rate, overall activity level of nearby neurons (which influences both cluster quality and SD-based spike thresholds), and correlation/synchrony (most sorters don't properly handle "collisions" between spikes in neighboring cells). Anesthesia could change any of these. Even though the physical configuration of the tetrode was the same, it is unclear how one could control for all of these issues when comparing two extracellular recordings.

5) Regarding Figure 6/Figure 7 directly linking the number of spikes evoked during a stimulus to the strength of inhibition/excitation during the stimulus is not entirely straightforward. Short-term synaptic plasticity is an equally important factor in determining a neuron's ability to affect firing in its targets. This could even be a mechanism for maintaining the supralinearity of inhibition; if inhibitory synapses don't depress as much as excitatory synapses do, they'll still be capable of stabilizing the network even when the network is very active, producing an inhibition-stabilized network in which the number of inhibitory spikes doesn't increase faster than the number of excitatory spikes (see e.g. Tan/Agmon 2008). In the converse case, in a network in which excitatory synapses depress less than inhibitory synapses, the network could appear inhibition-stabilized in its spike count and yet not be inhibition-stabilized in terms of the overall magnitude of the inhibitory conductances. Either way, STSP is a key factor and spike rates are only half the story. The question is whether Figure 6 really locks down the issue of whether cortical networks are actually inhibition-stabilized. It seems that the relationship between inhibition-stabilized networks and STSP is worth mentioning, even if only briefly.

---

## [Author Response]

Essential revisions:

*1) Both the chosen statistics and the qualitative comparisons between data and model (e.g. Figure 3) indicate very good agreement. However, one still wonders, how good can the fits be? And, what is the limiting factor, the dynamical repertoire of the model or the amount of data? These are important questions for determining not only the applicability of the model, but also what it would take to improve upon the current results.*

*Here is one way such a comparison could be made. The idea is to generate an estimate of the lowest possible error (or cost) given the data, and compare it to the actual error between model and data (for any given recording). To get such limit, divide the recorded data set into two equal parts, A and B (say, the first and second halves of the total recording time), and compare them using the same exact statistics and cost functions used to compare the model and the full data set, autocorrelation function, correlations, etc. If the total error between parts A and B is, say, 0.10, and that between model and data is, say, 0.20 (i.e., 100% larger), then we know that the model probably still has room to improve, i.e., one could fiddle with other parameters or make it more complex in order to generate a better match to the data. In contrast, if the error between model and data is, say, 0.11 (just 10% higher than that between parts A and B), it would suggest that the model is essentially matching the data as well as that can be done given the amount at hand, and the chosen statistics.*

*This, or something like this, would provide a more intuitive sense for how good the model is in relation to how good any model can be under the circumstances. It would indicate how drastic are the simplifications of the model. For instance, if the error between model and data is far higher than the bound found by randomization, it would suggest that less drastic simplifications would be able to capture more variance in the data.*

We have computed the upper limit for the explained variance of the model based on the recordings. We split each neural recording into two halves (interleaved segments of 4s each) and computed the amount of variance in the statistics from one half of the recording that is explained by the other half of the recording. The median variance explained for the autocorrelation function of the MUA, the distribution of MUA values across time bins, and the mean pairwise correlations were 84%, 98%, and 100% respectively. We compared this to the amount of variance in the statistics of the full recording that was explained by the model. The median variance explained for the three statistics were 82%, 90%, and 97% respectively. Thus, the model approaches the upper limit for the explained variance in both the autocorrelation function of the MUA and the mean pairwise correlations. The small amount of variance in the distribution of MUA values across time bins that is unexplained by the model is due primarily to outliers in the distribution in the recordings corresponding to bursting activity. We do not attempt to incorporate single-neuron bursting in our model and, thus, fail to explain these exceptionally high spike rates.

We have added a summary of this analysis in the main text and included the associated plots as Figure 3—figure supplement 3.

*2) The model would be much more useful if the results were more interpretable. Perhaps the authors could unpack one of the areas of simplification and explore how it affects the results. Here are two possibilities.*

*One, the findings would be more interpretable if they relied more on naturally occurring cortical states. Not only is the exact pharmacological action of the anesthetics still unclear, there is evidence that some drugs like ketamine act differentially on excitatory and inhibitory neurons, making interpretation of the model difficult. The model would provide a more powerful framework if the authors included recordings where a set of stable, identified neurons was followed across multiple quantified waking states (which could be distinguished by arousal state or EEG/LFP data), or from wakefulness to sleep.*

*Two, the authors assume that the activity of FS cells represents the state of inhibition in the cortex, and therefore conclude that the level of inhibition is low under some circumstances and higher in others. However, there are several inhibitory-inhibitory connections between different interneuron cell types (PV, SST, VIP etc.). The overall level of inhibition at any given moment may thus remain stable even as the contributing population of interneurons changes. Using a single feedback inhibitory mechanism may not be the most appropriate way to model inhibitory dynamics in the cortex.*

*For instance, one potential pitfall of incorporating a single feedback inhibition mechanism is that tuning might be sharpened simply by an 'iceberg' effect, where nonspecific increases in inhibition simply eliminate small responses to input. This was noted by Lee et al. 2013 and El-Boustani et al. 2014, following the Lee et al. 2012 paper cited by the authors. The authors should address this issue in interpreting the results from Figure 5.*

Regarding the first point, note that in the original manuscript our model was fit to both anesthetized (n = 44) and awake (n = 15) recordings. However, we did not originally include any awake recordings in which we were able to separate epochs corresponding to distinct behavioral states. In the revised manuscript, we now include analysis of new multi-neuron recordings from awake head-fixed mice that were able to run on a treadmill. It has been shown previously that locomotory behavior induces a more desynchronized cortical state (Polack et al., 2013), which we also verified in our new data. We separated these recordings into stationary and running epochs and asked which mechanisms in our model best explained the differences in cortical dynamics between these two behavioral states. The results were consistent with the rest of our analyses: we found that increases in inhibition best explained the change in dynamics, although an even better model of the transition between stationary and running periods was obtained by changes in both inhibition and adaptation. We also verified in these recordings that the relative increase in fast-spiking neuron activity that accompanied the desynchronization caused by running (87%) was significantly larger than the relative increase in regular-spiking activity (28%).

These new recordings and analysis are included in the revised manuscript as Figure 7.

Regarding the second point, we did not attempt in this study to address the contributions of different cortical inhibitory interneuron classes. Our analysis starts from the observation that feedback inhibition can be used to stabilize an otherwise oscillatory chaotic neural network. In real cortical networks, the overall level of feedback inhibition is likely to depend on multiple interneuron classes, perhaps with different interneuron classes playing more or less important roles in different situations. We found that the behavior of fast-spiking neurons in our data is at least consistent with a stabilization role in several situations, but our data do not enable us to assess the degree to which other interneuron classes may also be involved. We have revised the Discussion to propose the study of the roles of different interneuron classes in network stabilization as a topic for future work.

*3) There are several major inconsistencies between the experimental literature and model assumptions that should be discussed.*

*Regarding the claim that cortical activity is generally suppressed during active behaviors, all the papers cited in support of this claim are auditory (subsection “Neuromodulators and inhibitory control of cortical dynamics”). However, the exact opposite result is found in the visual system, in which cortical activity, RS response rate, SNR, and so on are all increased during locomotion. (There are numerous papers on this; Niell/Stryker 2010 is a particularly famous example.) This work doesn't fit with their theory at all! Rather than gloss over this issue, the authors should describe the relevant data both for and against their theory, and explain why their theory is still valuable even if not all its predictions are borne out.*

We were not clear enough on this point in the original manuscript and we have clarified it in the revised Discussion. We outlined two types of states with weak noise correlations and reliable evoked responses: the desynchronized, high spike rate states (as seen in Niell & Stryker, 2010 – mouse visual cortex) and desynchronized, hyperpolarized states (as seen in McGinley et al., 2015 – mouse auditory cortex). There are also several studies that report little or no change in spike rate in pyramidal neurons visual cortex during running (Dipoppa et al., 2016; Pakan et al., 2016; Polack et al., 2013; Bennett et al., 2013, all in mouse visual cortex L2/3).

In the revised manuscript, we now also report on our own recordings made in mouse visual cortex during running as described above. We find that running tends to increase spike rates in both RS and FS neurons relative to periods in which the animal is stationary. In the new Figure 7 of our revised manuscript, we show that increased inhibition is still necessary to stabilize the desynchronized network and decrease correlations during running, even when there is an increase in spike rate, consistent with our other results. Because of the increase in spike rate, another parameter must also change in addition to inhibition, such as adaptation, in order to explain the increase in RS spiking activity (note FS activity increases in 4/4 of our recordings and RS activity increases in 3/4). Note that these results are similar to those reported for the transition from synchronized to desynchronized states under urethane in the original manuscript. Desynchronization under urethane is similar to desynchronization during running in that the spike rates of both FS and RS neurons increase, with the increase in FS activity being proportionally larger. In the case of urethane, the increase in RS activity despite increased FS activity appears to be due to an increase in tonic input.

Thus, in summary, our modelling results are not inconsistent with the desynchronized high spike rate state; instead, our results suggest that another parameter in addition to inhibition must also change in order to fully capture the transition to a stabilized network with higher spike rates.

*Another example is their suggestion that many FS cells do not respond to stimuli under ketamine anesthesia, and that this silencing is responsible for the tendency of cortical networks to generate Up/Down dynamics under ketamine. The issue of whether FS cells are responsive under ketamine was examined by Moore/Wehr 2013 (who identified PV cells using optotagging in mouse auditory cortex under ketamine anesthesia, and showed that most tagged cells had high spontaneous and evoked firing rates), as well as Cohen/Mizrahi 2013 (who randomly patched neurons in ketamine-anesthetized rat barrel cortex, identified the FS neurons by intracellular current injection, and showed that all FS cells did respond either to ipsi- or contra-lateral whisker stimulation). How is the model consistent with these data – or if it's not, can the authors discuss why the model is nonetheless worth considering?*

Our assertion in the original manuscript that many FS cells do not respond to stimuli under ketamine was based on weak evidence. A well-designed experiment to address this issue would require recordings in which the same neurons were studied under ketamine anesthesia and in another state, which we do not have. Our attempt to compare recordings from different populations that were recorded under ketamine and under fentanyl was not sufficient; the observation that our recordings contained fewer FS cells under ketamine could have many explanations (as pointed out by the reviewer in point 4 below) and, thus, is not a sufficient basis for asserting that many FS cells go silent under ketamine. We have removed this assertion and the associated analysis from the revised manuscript.

Regarding the question of whether existing data support the results of our model-based analysis which suggests that ketamine induces a state of weak inhibition: There are a number of previous studies suggesting that NMDA antagonists such as ketamine disinhibit cortex (Olney et al., 1999). This disinhibition may occur through a reduction in inhibitory activity: ketamine has been shown to reduce the immunoreactivity of GABAergic neurons in cortical slices (Kinney et al., 2006; Behrens et al., 2007). The Moore/Wehr 2013 study does not really speak directly to this question. In that study, the authors made no attempt to make an unbiased measurement of the overall level of inhibitory activity in the population under ketamine relative to any other state (or, indeed, at all). The same is true of Cohen/Mizrahi 2013: they may have randomly patched neurons, but they pre-selected cells for inclusion in their study based on a number of very specific criteria. This is also true of the more recent study from the same lab (Maor et al., Cer Cortex, 2016) in which they made targeted recordings from PV+ cells with visual guidance, but included only those cells with evoked responses in their analysis. So, in summary, neither existing data nor our own provide a reliable measurement of the overall level of inhibitory activity under ketamine relative to other states.

*Regarding waveform classification (subsection “Classifying FS and RS neurons”): There is a genuine debate in the literature regarding the relationship between cell type and extracellularly recorded spike waveform, but one wouldn't get that impression from reading this paragraph: first, because only one paper is cited on the "exercise caution" side, and second, because no papers at all are cited to support the preferred estimate of 90-100% correct classification. It would be much better to cite papers on both sides of the debate, since this classification is so central to the analysis.*

The claim of 90-100% correct classification was made both in the main text and in the Methods, but the references supporting this claim were only cited in the main text. We have now duplicated this list of references in the Methods. We have also added an additional study that provides very strong evidence for the validity of waveform-based classification (Cohen and Mizrahi, 2015): In Figure 1 of their paper, these authors show using loose-patch recordings in mouse auditory cortex of PV-cre-tdTomato mice that 71/73 tdTomato+ cells had a peak to valley duration < 1ms (equivalent to trough-to-peak in our recordings), and 70/74 tdTomato- cells had a peak to valley duration > 1ms.

There are also 5 other independent studies which all quantitatively support the estimate of 90-100% correct classification: Madisen et al., 2012 (Figure 7), Cardin et al., 2009 (Figure 2), Cho et al., 2010 (Figure 3), Bartho et al., 2004 (Table 1), and Stark et al., 2013 (Figure 1).

The evidence against the validity of waveform-based classification seems to be based largely on the observation of a particular, rare, class of pyramidal neurons with narrow spikes called “chattering cells”, which have only been shown to exist in carnivores, and have not yet been found in rodents or primates, as far as we know (Constantinople et al., 2010). Even in carnivores, these cells only occur in layers 2/3 (Gray and McCormick, 1996).

We cited the only study in rodents that we are aware of that is apparently inconsistent with the validity of spike width-based classification (Moore and Wehr, 2013). In our revised Methods, we discuss the reasons why their results were likely due to the specifics of their analysis methods. First, Moore and Wehr perform their analysis on filtered spike waveforms, which might explain why all of their RS neurons have upward-going spikes (their Figure 1). We only observe such spikes in less than 1% of the population throughout all our cortical recordings. Second, they used a measure of spike width that is different from ours. Like the studies we cite above, we used trough-to-peak duration as a measure of spike width, while Moore and Wehr use the entire duration of the spike, based on custom defined thresholds. Their Figure 1 suggests that their classification would have been more successful if they had used trough-to-peak. Moreover, Moore and Wehr also did not have a bimodal distribution of spike widths after filtering, which we reliably find in all 70+ of our recordings for trough-to-peak distributions.

*Regarding whether the FS/RS distinction "is still effective for approximating the overall levels of inhibitory and excitatory activity in a population" (subsection “Activity of fast-spiking (FS) neurons is increased during periods of cortical desynchronization with weak noise correlations”): It is not necessarily true that activity of all types of inhibitory cell will be correlated with one another, particularly since inhibitory cells can inhibit one another in targeted, type-specific ways (e.g. Pfeffer/Scanziani 2013), which will tend to produce anticorrelations between activity in the different classes of interneurons. Fanselow/Connors 2008 showed that the states in which somatostatin cells were active tended to be those in which parvalbumin cells were least active, and in fact Tan/Agmon 2008 suggest that non-FS inhibitory cells (somatostatin cells) are the ones most specialized to provide the kind of stabilizing influence the authors' models require.*

We were not clear enough about this issue in the original manuscript. We do not know that PV+ cells alone provide the stabilizing effect on the network’s activity. We propose that any increase in inhibitory feedback would reduce correlations, which could include SST neurons (Tan & Agmon, 2008). Unfortunately, in our recordings we do not have the ability to identify SST cells and track their activity to determine if their activity increases with decreased correlations. We have clarified this point in the revised Discussion.

Regarding Fanselow/Connors 2008: they showed two in vitro states in which GIN (subtype of somatostatin) cells were more active than FS cells (application of ACSF and application of an mGluR agonist DHPG). Under these preparations, there was no FS spiking at all, which is inconsistent with all current in vivo studies of PV cells in awake mice that we know of, suggesting that the states induced by these preparations may not be physiologically relevant. There may indeed be a decrease in PV activity when SST activity increases in certain situations, but we are not aware of any in vivo studies that report the size of this effect.

*4) Subsection “Activity of fast-spiking (FS) neurons is increased during periods of cortical desynchronization with weak noise correlations”, last paragraph/Figure 7 shows differences in the total numbers of thin versus broad spikes under fentanyl versus ketamine anesthesia. Many factors can influence how many units/clusters are recovered from a recording and how well spikes are assigned to their underlying sources, including physical/pulsatile instability of the brain caused by breathing and heart rate, overall activity level of nearby neurons (which influences both cluster quality and SD-based spike thresholds), and correlation/synchrony (most sorters don't properly handle "collisions" between spikes in neighboring cells). Anesthesia could change any of these. Even though the physical configuration of the tetrode was the same, it is unclear how one could control for all of these issues when comparing two extracellular recordings.*

We agree that this analysis is problematic for the reasons described above and we have removed it from the manuscript.

*5) Regarding Figure 6, directly linking the number of spikes evoked during a stimulus to the strength of inhibition/excitation during the stimulus is not entirely straightforward. Short-term synaptic plasticity is an equally important factor in determining a neuron's ability to affect firing in its targets. This could even be a mechanism for maintaining the supralinearity of inhibition; if inhibitory synapses don't depress as much as excitatory synapses do, they'll still be capable of stabilizing the network even when the network is very active, producing an inhibition-stabilized network in which the number of inhibitory spikes doesn't increase faster than the number of excitatory spikes (see e.g. Tan/Agmon 2008). In the converse case, in a network in which excitatory synapses depress less than inhibitory synapses, the network could appear inhibition-stabilized in its spike count and yet not be inhibition-stabilized in terms of the overall magnitude of the inhibitory conductances. Either way, STSP is a key factor and spike rates are only half the story. The question is whether Figure 6 really locks down the issue of whether cortical networks are actually inhibition-stabilized. It seems that the relationship between inhibition-stabilized networks and STSP is worth mentioning, even if only briefly.*

We have added a brief discussion of STSP in the revised manuscript. Here we provide a more detailed response to this comment:

Several previous studies (Tsodyks et al., 1998, Loebel et al., 2007) have modelled slow-timescale up/down state transitions using synaptic depression rather than spike frequency adaptation. We chose not to include such short-term synaptic plasticity in our model so as to limit the number of parameters. But in a simplistic EI model, none of the potentially adapting synapses would have the effects that we observed on network activity and FS interneurons. The first possibility suggested by the reviewers is that inhibitory synapses might not depress as much as excitatory synapses. If this was true, then during desynchronization, the spike rates of inhibitory neurons would increase/decrease together with those of excitatory neurons, while the IE synapses would dominate over EE synapses, thus stabilizing the network. But in all cases, we observed that desynchronization caused a relative increase in FS activity that was larger than the corresponding relative increase in RS activity. The second possibility suggested by the reviewers is that excitatory synapses depress less than inhibitory synapses, thereby making the activity of inhibitory neurons large, even though their synapses are weak and unable to stabilize the dynamics. But we observe the opposite: the dynamics appear to be most stabilized in our recordings when FS activity is particularly high (desynchronized states).

So we concluded that introducing synaptic depression into our model was unlikely to significantly modify our conclusions, and to model it properly we would need multiple adaptation parameters on E and I synapses, which we did not think we could estimate robustly together with our other 5 parameters.